# Day-Ahead Scheduling Model of the Distributed Small Hydro-Wind-Energy Storage Power System Based on Two-Stage Stochastic Robust Optimization

**Jun Dong, Peiwen Yang and Shilin Nie \***

School of Economics and Management, North China Electric Power University, Beijing 102206, China;
dongjun@ncepu.edu.cn (J.D.); 1172206214@ncepu.edu.cn (P.Y.)
**\*** Correspondence: 1182206131@ncepu.edu.cn

**Abstract:** With renewable energy sources (RESs) highly penetrating into the power system, new problems emerge for the independent system operator (ISO) to maintain and keep the power system safe and reliable in the day-ahead dispatching process under the fluctuation caused by renewable energy. In this paper, considering the small hydropower with no reservoir, different from the other hydro optimization research and wind power uncertain circumstances, a day-ahead scheduling model is proposed for a distributed power grid system which contains several distributed generators, such as small hydropower and wind power, and energy storage systems. To solve this model, a two-stage stochastic robust optimization approach is presented to smooth out hydro power and wind power output fluctuation with the aim of minimizing the total expected system operation cost under multiple cluster water inflow scenarios, and the worst case of wind power output uncertainty. More specifically, before dispatching and clearing, it is necessary to cluster the historical inflow scenarios of small hydropower into several typical scenarios via the Fuzzy C-means (FCM) clustering method, and then the clustering comprehensive quality (CCQ) method is also presented to evaluate whether these scenarios are representative, which has previously been ignored by cluster research. It can be found through numerical examples that FCM-CCQ can explain the classification more reasonably than the common clustering method. Then we optimize the two stage scheduling, which contain the pre-clearing stage and the rescheduling stage under each typical inflow scenario after clustering, and then calculate the final operating cost under the worst wind power output scenario. To conduct the proposed model, the day-ahead scheduling procedure on the Institute of Electrical and Electronics Engineers (IEEE) 30-bus test system is simulated with real hydropower and wind power data. Compared with traditional deterministic optimization, the results of two-stage stochastic robust optimization structured in this paper, increases the total cost of the system, but enhances the conservative scheduling strategy, improves the stability and reliability of the power system, and reduces the risk of decision-making simultaneously.

**Keywords:** distributed power system; high renewable penetration; fuzzy C-means clustering and clustering comprehensive quality method; two-stage stochastic robust optimization

---

## 1. Introduction

Since recently, the total amount of fossil energy has been decreasing gradually, and global environmental pollution is becoming increasingly serious. Reducing carbon emissions and improving energy efficiency is the top priority [1,2]. Moreover, the expected incoming energy crisis makes people pay more attention to renewable energy for its cleanness, technology maturity and renewability.

For these reasons, the ratio of renewable energy sources (RESs) generation has been growing rapidly worldwide. Hydropower, as an important form of renewable energy generation, is one of

the major contributors to the future power supply. In China, the total installed power capacity of hydropower in 2017 ranked second only to thermal power. This further indicates that the development of hydropower in China is gradually valued.

Sichuan province is rich in water resources, ranking first in the country. The province's average annual precipitation is about 488.975 billion cubic meters. The water resource is most abundant in river runoff, with nearly 1400 rivers of various sizes within the territory, and the total water resource is about 348.97 billion cubic meters. Therefore, the proportion of hydropower installed in Sichuan ranks first in China, accounting for about 80% of the total installed power in the whole province [3]. It is precisely because Sichuan has many rivers, and the catchment area is relatively dispersed, that for a large number of small hydropower plants, most of which are run-of-river, are distributed in it. These small hydropower stations have few reservoirs, so it is virtually impossible to store water, which means that the flow of water either generates electricity or is spilled, depending on the power capacity of the small power plants [4]. However, there are still existing some problems in the distributed small hydropower plants (SHPPs) in Sichuan province; for instance, unordered power generation plans, poor coordination between SHPPs, and lack of centralized control. Meanwhile, the development of wind power in Sichuan is relatively rapid, and how to effectively utilize these distributed wind power plants (WPPs) and SHPPs is the key to promoting the development of renewable energy in Sichuan.

In order to better absorb the distributed energy sources, it is a better choice to trade them in the day-ahead power market. However, the smooth implementation of day-ahead scheduling is the guarantee of distributed energy transaction. By consulting the relevant historical literatures on hydropower dispatching, it can be found that numerous studies have focused on the joint operation of hydropower stations and new energy [5]. Yue Chen et al. [6] presents a distributionally robust hydro-thermal-wind economic dispatch, and it is notable that the hydropower units in this reference retain some spinning reserve capacity, namely, that they are capable of self-regulation. Besides, the hydropower plants mentioned in [7–12] are commonly equipped with reservoirs, which can store water energy, and adjust the output of hydropower stations according to the power generation demand. In summary, most of the hydropower stations mentioned in the literature on hydropower dispatching have some adjustability and flexibility due to the existence of reservoirs [7]. Thus, part of the capacity of hydropower stations can be retained for spinning reserve [8], which is why hydro and wind can synergistically operate with each other [12].

However, not all hydropower plants have reservoirs, nor are they adjustable [13], but there are also many run-of-river small hydropower plants, which have no storage capacity. The power generation of these plants is restricted by the natural incoming water, and the capacity of the power plants, which has significant seasonal characteristics [14]. Consequently, apart from the different forms of power generation, the characteristics of power generation and wind power are essentially the same in these run-of-river SHPPs. When the ISO conducts the scheduling for the power system that accesses to distributed WPPs and SHPPs simultaneously, how to handle the uncertainty of these two renewable sources is the key issue. Several methods have been presented to solve the day-ahead economic dispatch (DAED), considering the uncertainty of renewable energy. In [15], applying the cumulative density function (CDF) and probability density function (PDF) of wind power describes the probability distribution of its power generation. In addition, [16] proposes a scenarios generation method based on Markov chains to generate a predicted output of wind power with several scenarios, and analyzes the influence of wind fluctuation on power system operation under different wind power output scenarios. Besides, some literatures adopt a scenario clustering method instead of a scenario generation method to describe the uncertainty of RESs. For instance, [17] uses k-means clustering and the discrete-time Markov chain (DTMC) synthesis method to cluster a typical solar radiation curve. [18] proposes a centroid-based clustering algorithm to calculate the whole portfolio flexibility. Compared to the k-means, [19] proposed the fuzzy k-means, which is much better for the clustering results.

In addition to this, [20] compares several clustering methods, and finds that k-shape improves performance significantly over traditional clustering methods. Compared with k-means and

fuzzy-c-means, this k-shape, or some other advanced clustering methods can improve the efficiency of clustering. Even so, most clustering methods still have a common disadvantage that they ignore the evaluation of the final clustering results. In other words, the clustering methods generally give a certain number of clustering before, but the classification is not estimated, which is the lack of such research at present.

Through summarizing the literature, it can be found that there are still some problems remaining to do with dispatching a distributed power system. Hence, the day-ahead scheduling problem of a distributed power system studied in this paper not only considers wind power, but also the run-of-river SHPPs, and introduces thermal power and energy storage devices, which control the uncertainty of small hydropower and wind power by utilizing the charging and discharging process of energy storage and the reserve capacity of thermal power. Moreover, a two-stage stochastic robust optimization model, which is the combination of robust optimization [21] and stochastic optimization, and seldom used in the study of this kind of problem, is proposed, and used to deal with the uncertainty of small hydropower and wind power, respectively, with the idea of stochastic optimization and robust optimization. In the aspect of scene clustering, the FCM-CCQ algorithm, which can not only cluster quickly and effectively, but also evaluate the quality of clustering results, is adopted to cluster the historical upstream inflow data of small hydropower, and can reasonably explain why the clustering scheme is the best one. Overall, compared with the previous research of power system dispatching, the main contributions of this paper are threefold:

(1) A day-ahead scheduling model, accompanied with a high penetration of RESs, is adopted for the distributed power system. Specifically, distinguishing from most hydropower plants in other literatures, the small hydropower plants in this paper are run-of-river, and distributed without regulating power capacity. To operate and regulate this power system steadily, it is crucial that the ISO should reserve sufficient spare capacity to smooth out the RESs' volatility, and follow the grid security constraints. Thus, the participants of the battery energy storage system (BESS) and thermal power plants (TPPs) are particularly important.

(2) Based on the Fuzzy C-means (FCM) clustering method, the number of historical scenarios of small hydropower water inflow can reasonably reduce to a fixed value. Then, a novel clustering evaluation method named clustering comprehensive quality (CCQ) is proposed to determine the number of best scenario categories. The typical scenarios are determined by the expected value of scenarios within the same category, and the probability distribution of each typical scenario is finally determined to provide support for subsequent models to get the expected cost.

(3) A two-stage adaptive robust optimization based on several scenarios, settled is proposed for applying to day-ahead scheduling process, which takes the minimum expected operation cost of the whole power system, considering the RESs' fluctuations as the optimization objective. The solving method of this optimization is a column-and-constraint generation (CCG) algorithm, and using the alternating direction (AD) algorithm to deal with the inner level problem.

The rest of the paper is organized as follows: Section 2 shows the main research problem description. Section 3 presents the objective functions and constraints of the day-ahead scheduling model of the distributed small hydro-wind-energy storage power system, and the solution of the proposed model. Section 4 simulates a case study of the day-ahead clearing and analyzes the optimal results. Section 5 summarizes the main conclusions in this paper.

## 2. Problem Description

In this section, the main problem is briefly summarized in this paper, which contains the participants of the power system, and the proposed scheduling process.

### 2.1. Power System Participants

In this study, the participants of this power system consist of renewable distributed generators (RDGs) (such as SHPPs and WPPs), conventional generators (TPPs), energy storage system (BESSs) and electric loads. Renewable distributed generators can provide emission-free and sustainable energy, but the volatility of power output remains their most obvious drawback. TPP, as a conventional generator set, has some flexibility, and can help maintain power balance and the stability of the entire power system. However, due to its large environmental pollution, the unit is used in a lower order. BESS can decrease surplus water and wind curtailment by flexibly switching charge/discharge states to transfer electrical energy. Because BESS is faster than the frequency modulation rate of thermal power, using it to coordinate water and wind power generation in the real-time stage will reduce the impact on the grid security. Electricity loads are assumed to be an inelastic fixed predicted value in any bus. The architecture of the distributed power system in our case is demonstrated in Figure 1.

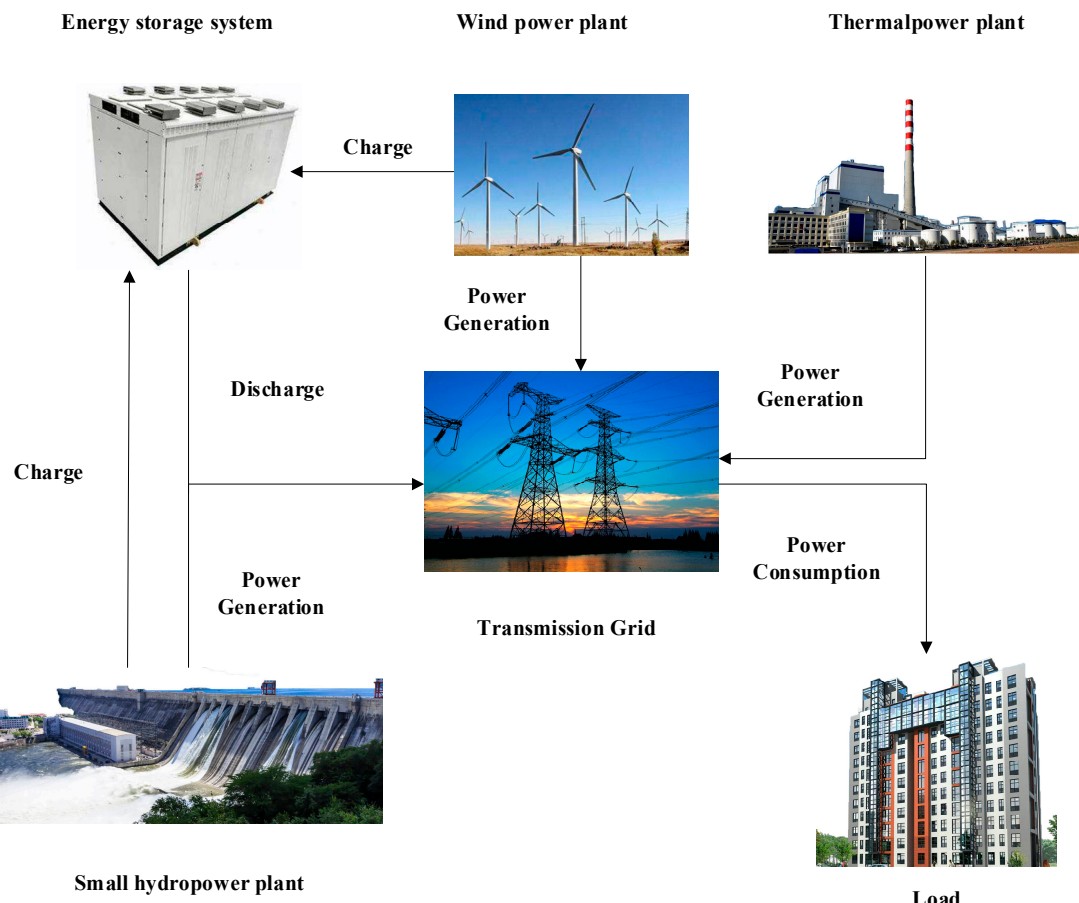

**Figure 1.** The structure of a power system.

### 2.2. Proposed Scheduling Process

In this paper, a scheduling process in day-ahead marketing is put forward for a distributed power system with high RESs penetration. The overall goal of scheduling focus on minimizing the total operation cost of the whole power system.

Specifically, the scheduling can be divided into two stages. At the first one, according to the possible water supply scenarios of small hydropower, and the predicted output range of wind power, ISO arranges scheduled generation under the condition that the power flow constraints are met. In this stage, the problem is modeled as a deterministic optimization without taking into account the uncertainty in the power system. At the second stage, this involves simulating the real-time power dispatching process with the objective of considering the minimum operation cost of real-time

scheduling. TPPs and BESSs, as power balance units, make use of the flexibility of power modulation to adjust up or down, so as to equilibrate the power deviation caused by inaccurate predictions of hydropower and wind power. In particular, BESSs can be regarded as a power pool to store the wind power and hydropower generation, save the excess electricity, and release it when the power generation is insufficient. Through the charging and discharging process of BESSs, it curtails the damage of wind power and hydropower output volatility to the power grid.

## 3. Model Construction and Solution

In this section, problem assumptions should be given before the model construction. Then, the definition of the uncertainty sets is given first, followed by the two-stage stochastic robust optimal dispatching model. Next, reformulate the scheduling model, and present the solution methodology at last.

### 3.1. Problem Assumptions

In real life, day-ahead market scheduling optimization is a very complicated process, which is accompanied by the dynamic change of market members' quotation strategy and complex network blocking constraints. In this paper, for the sake of simplicity, and to convey the core theme of this paper, we make the following assumptions:

(1) Both security constraints unit-commitment (SCUC) are not considered. The default unit-commitment has been optimized before studying this problem.
(2) All SHPPs in the distributed system in this paper are run-of-river hydropower stations.
(3) The network power flow constraint is treated as DC by default, and any transmission line loss is not taken into account.
(4) The load demand in the distributed system is processed according to the predicted value, without considering the uncertainty of the load side and the demand response (DR).
(5) It is assumed that ISO can accurately predict the output interval of WPP in each period of the next day, based on the historical data of the wind turbine at a certain confidence level.

### 3.2. Definition of the Uncertainty Sets

In order to surmount the uncertainty of SHPPs and WPPs, the uncertainty should be simulated so as to support the ISO for decision making in day-ahead scheduling [22]. More details about the handling method are in the following article.

3.2.1. SHPPs Uncertainty Sets

To decrease the number of model scenarios and establish the probability distribution of future small hydropower inflow, scenario reduction should be carried out, based on the historical original scene set. Clustering is an effective method for scenario reduction. This paper proposes a scenario reduction strategy for small hydropower inflow based on FCM and CCQ. In detail, the FCM clustering method is used to realize scenario division of similar categories and the CCQ method is used to regulate the optimal number of scenario categories. In the same category [23], the typical scenario is formulated by the situational expectation, and then the probability distribution of each scenario is calculated by the membership matrix. The clustering and evaluation methods used in this paper are described in detail below.

(1) FCM clustering method

According to the different characteristics and similarity of objective things, this method is a mathematical method to classify them by establishing fuzzy similarity relations. Derived from previous research results [24], this paper takes the minimum clustering losses based on membership functions. The formulation of the objective function is shown as follows:

$$J(U, V, W) = \min\left(\sum_{j=1}^{n} \sum_{k=1}^{c} U_{kj}^{b} D_{kj}^{2}\right) \tag{1}$$

$$\begin{cases} D_{kj} = \sqrt{\sum\limits_{i=1}^{m} \left[w_i \big| x_{ij} - v_{ij} \big|\right]^2} \\ X = \{x_1, x_2, \ldots x_n\} \\ \sum\limits_{k=1}^{c} U_{kj} = 1, 0 \le U_{kj} \le 1, \sum\limits_{j=1}^{n} U_{kj} > 0 \\ \sum\limits_{i=1}^{m} w_i = 1, 0 \le w_i \le 1 \end{cases} \tag{2}$$

$$\sum_{k=1}^{c} U_{kj} = 1 \tag{3}$$

$$0 \le U_{kj} \le 1 \tag{4}$$

$$\sum_{j=1}^{n} U_{kj} > 0 \tag{5}$$

where $n$ denotes the total number of scenarios; $c$ is the clustering number; $b$ refers to the weighted index, also known as the smoothing factor, which controls the degree of sharing between fuzzy categories. There is no theoretical guidance on the optimal value of $b$, which is 2 in most cases. $U_{kj}$ denotes the relative membership of sample $j$ belong to category $k$, and Equations (3)–(5) are the constraints of $U_{kj}$; $D_{kj}$ refers to the Euclidean distance between scenario $j$ and the central of category $k$; $X$ represents the sample set; $m$ is the number of index eigenvalues; $w_i$ denotes the weight of the $i$-th index, and equal weight is adopted in this paper ($w_i = \frac{1}{m}$); $x_{ij}$ refers to the $i$-th eigenvalues in scenario $j$; $v_{ij}$ is the standardized eigenvalue $i$ of category k central, which is between 0 and 1.

Generally, since there are different degrees of difference in the dimension and magnitude of each indicator, if directly calculated, it may increase the effect of some characteristic indices with a large magnitude on the classification, and reduce the effect of some small-scales, thereby changing the classification result. Therefore, the eigenvalues of each indicator need to be normalized, and then $D_{kj}$ can be reformulated as:

$$D_{kj} = \sqrt{\sum_{i=1}^{m} \left[ \frac{1}{m} \left| \frac{x_{ij} - x_{ij}^{\min}}{x_{ij}^{\max} - x_{ij}^{\min}} \right| - v_{ik} \right]^2} \tag{6}$$

where $a_{ij}^{\max}$, $a_{ij}^{\min}$ represent the maximum and minimum eigenvalues of the $i$-th indicator in the scenario set, respectively.

By constructing the Lagrangian function to solve the extreme value problem of Equation (1), satisfying the constraints (2)–(5), the clustering iterative Equations (7) and (8) can be obtained. After cyclic iteration until convergence conditions are met, scenario clustering is finally realized.

$$U_{kj} = \frac{1}{\sum\limits_{l=1}^{c} \left[ \frac{\sum\limits_{i=1}^{m} \left( \frac{x_{ij} - x_{ij}^{\min}}{x_{ij}^{\max} - x_{ij}^{\min}} - v_{ik} \right)^2}{\sum\limits_{i=1}^{m} \left( \frac{x_{ij} - x_{ij}^{\min}}{x_{ij}^{\max} - x_{ij}^{\min}} - v_{il} \right)^2} \right]} \tag{7}$$

$$v_{ik} = \frac{\sum\limits_{j=1}^{n} U_{kj}^{2} \frac{x_{ij} - x_{ij}^{\min}}{x_{ij}^{\max} - x_{ij}^{\min}}}{\sum\limits_{j=1}^{n} U_{kj}^{2}} \tag{8}$$

(2)　CCQ method

The number of appropriate clustering scenarios can accurately reflect the features of SHPP output, and has a significant impact on the solution effect of the model.

In general, the natural attributes of the clustering result distribution are used to evaluate the identity within the cluster and the separation between clusters, which is consistent with the clustering goal of maximizing intra-cluster similarity, and minimizing the similarity between clusters. Therefore, clustering density and proximity are often used for cluster evaluation, and the determination of the optimal number of clusters. In this paper, the CCQ method combining the above two properties is adopted to determine the best scenario category.

Clustering density is measured by intra-cluster variance. Given a data set $X$ corresponding to an eigenvalue $i$, the variance can be formulized as:

$$\text{var}(X) = \sqrt{\frac{1}{n}\sum_{j=1}^{n}(x_j - \overline{x})^2} \tag{9}$$

where $X$ is the total scenarios of set; $\overline{x}$ denotes the mean value of a sample corresponding to one of the eigenvalue; $x_j$ represents the value of this indicator in scenario $j$.

As for the clustering density of the results $C_1, C_2, C_3 \ldots C_C$, it can be calculated by the following formula

$$D_{en} = \frac{1}{C}\sum_{j=1}^{C}\left[\frac{\text{var}(c_j)}{\text{var}(X)}\right] \tag{10}$$

where $C$ is the sum scenarios after clustering; $\text{var}(c_j)$ represents the variance in category $C_j$. The samples in each cluster should be as close as possible, so the smaller the clustering density, the better.

The clustering proximity is defined as follows:

$$p_{ro} = \frac{1}{C(C-1)}\sum_{j=1}^{C}\sum_{k=1,k\neq j}^{C}\exp\left[\frac{-(S_{C_j} - S_{C_k})^2}{2\delta^2}\right] \tag{11}$$

where $S_{C_j}$ is the clustering center in category $C_j$; $\delta$ is Gaussian constant, and $2\delta^2 = 1$ for simplified calculation. The clustering proximity is inversely proportional to the distance between clustering. The smaller the clustering proximity, the more effectively the clustering can be divided, i.e., a better classification effect.

$$C_{om} = 1 - [\xi D_{en} + (1 - \xi)p_{ro}] \tag{12}$$

where $\xi \in [0, 1]$ is the weight of equilibrium clustering density and clustering proximity, which is 0.5 in this article. The average comprehensive quality evaluation of each index $\overline{C}_{om}$ determines the optimal number of clustering, as shown in the following formula:

$$\overline{C}_{om} = \frac{1}{m}\sum_{i=1}^{m}C_{om} \tag{13}$$

With the increase of the number of clustering, it is helpful for the average comprehensive quality of clustering associated with a dramatic enhancement to capture more detailed data objects, reduce the internal variance of clustering effectively, and separate the categories accurately. However, if the number of clusters is too large, the edge effect of increasing the average cluster quality may decrease, that is, splitting one cluster into two causes only a slight increase in the cluster quality. With the diversity diverse of the clustering scene number $C$, the $\overline{C}_{om}$ is also different, that is, each classification scheme corresponds to one $\overline{C}_{om}$. With this in mind, we can get a curve about the average comprehensive

quality of clustering and the number of clusters. Thus, choosing the right number of clusters by the heuristic method, which is the inflection point of this curve.

In our case, the sample set $Q = \{Q_1, Q_2, Q_3, \ldots, Q_n\}$ with a total number of $n$ is selected from the historical water inflow data, and the value of $C$ is assigned. As shown in Figure 2, the FCM method is proposed for clustering analysis and evaluating the quality of each classification, and the optimal number of clustering scenes is finally determined. In terms of the membership matrix, the water quantity and probability distribution of each typical scene can be obtained.

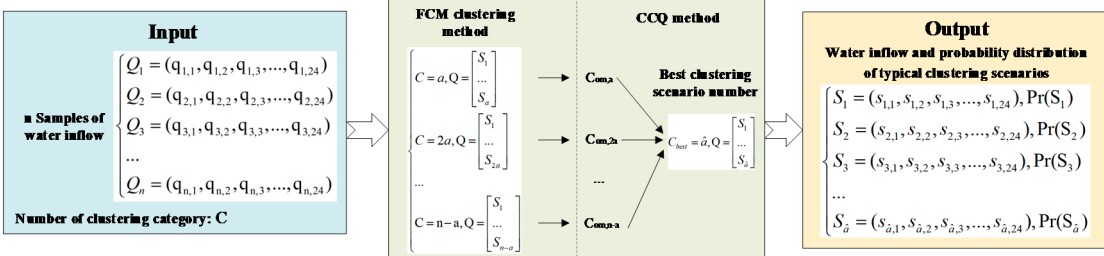

**Figure 2.** The water inflow scene clustering process of SHPPs.

### 3.2.2. WPPs Uncertainty Sets

WPPs real-time output cannot be predicted with great accuracy during the scheduling horizon, which drives ISO to show great concern about the presence of WPPs' output uncertainty when enforcing the scheduling. However, ISO can accurately forecast the real-time WPPs' output interval of each of the WPPs, based on its previous data of real-time WPP output. Assuming that there are NW WPPs in a distributed power system, the uncertainty set which corresponds to these WPPs' real time power output can be expressed as [25,26]:

$$
\Omega := \left\{
\begin{array}{l}
\widehat{P}_t^W = \left( \widehat{P}_{1,t}^W, \widehat{P}_{2,t}^W, \widehat{P}_{3,t}^W, \ldots \widehat{P}_{k,t}^W, \ldots \widehat{P}_{NW,t}^W \right): \\
P_{k,\min}^W \leq \widehat{P}_{k,t}^W \leq P_{k,\max}^W, \forall k, t, \\
\displaystyle\sum_{k=1}^{NW} \frac{\left| 2\widehat{P}_{k,t}^W - (P_{k,\min}^W + P_{i,\max}^W) \right|}{P_{k,\max}^W - P_{k,\min}^W} \leq \Gamma \sqrt{NW}
\end{array}
\right\}
\tag{14}
$$

where $\widehat{P}_{k,t}^W$ represents the real-time power output of the $k$-th WPP at stage $t$; $P_{k,\min}^W$, $P_{i,\max}^W$ are the lower upper predicted output confidence interval of WPP at time $t$; $NW$ denotes the number of WPPs; $\Gamma$ refers to the budget parameter which controls the size of the uncertainty set.

### 3.3. Stochastic Robust Optimal Dispatching Model

According to the description of the scheduling scenario in Section 2.2, we present a stochastic robust scheduling model, which mainly focus on the economic scheduling process. Thus, the security constrained unit-commitment issue is beyond the scope of this study. Stochastic robust optimization is an optimization method based on the uncertainty of influencing factors. Therefore, when adopting stochastic robust optimization to carry out day-ahead scheduling, ISO needs to ensure the worst impact of uncertain factors upon the system operation cost. Although the worst case may not occur in reality, ISO still needs to pay attention to it, and make decisions accordingly, so as to avoid the serious consequences brought on by the worst case. In this paper, we handle the randomicity of small hydropower inflows by scene clustering, and adopt the idea of robust optimization to address the uncertainty caused by wind power. In this model, the ultimate goal is to calculate the minimum expected cost of a day-ahead scheduling operation.

To minimize the overall expected operating costs of the distributed power system, the objective function is shown as Equation (15). The objective function of this problem is similar to most previous studies on ED problems, which consists of four terms: The operation cost of SHPPs, TPPs, WPPs, and BESSs, respectively.

$$\min f = \sum_{t=1}^{T} \left( \sum_{i=1}^{NH} \sum_{s=1}^{C} \rho_s C_i P_{i,s,t}^{H} \Delta t + \sum_{j=1}^{NT} C_j P_{j,t}^{T} \Delta t + \sum_{k=1}^{NW} C_k P_{k,t}^{W} \Delta t + \sum_{l}^{NB} C_{l,t}^{BESS} \right) \tag{15}$$

where $\rho_s$ denotes the probability of water inflow scenario s; $P_{i,s,t}^{H}$ is the output of SHPP $i$ in scenario s at time period $t$; $P_{j,t}^{T}$ represents TPP $j$ power generation at time period $t$; $P_{k,t}^{W}$ is the output of WPP $k$ at time period $t$; $E_{l,t}$ is the stored energy in BESS at time period $t$; represent $NH, NT, NW, NB$ the number of SHPP, TPP, WPP and BESS respectively; $t$ is the index of time; $\sigma_l$ refers to a leakage loss factor of the BESS. For the sake of simplicity, the cost of TPP per unit $C_j$ is regarded as a constant; $C_i, C_k$ are cost coefficient per unit of electricity of SHPP and WPP, separately, which can be calculated as below [27]:

$$C_i = \frac{r I_i^{ph}}{\left[ 1 - (1+r)^{-LC^H} \right]} + O_i^{ph} \tag{16}$$

$$C_k = \frac{r I_k^{pw}}{\left[ 1 - (1+r)^{-LC^W} \right]} + O_k^{pw} \tag{17}$$

$r$ is the discount ratio; $I_i^{ph}$, $I_k^{pw}$ represent the initial investment cost of the *i*-th SHPP and the *k*-th WPP; $O_i^{ph}$, $O_k^{pw}$ denote the operational and maintenance cost of the *i*-th SHPP and the *k*-th WPP; $LC^H$, $LC^W$, represent the life cycle of SHPP and WPP.

### 3.3.1. SHPP

Unlike the conventional hydropower plant, SHPP is in most instances "run-of-river"; in other words, it has almost no water storage function, and even if it does have a reservoir, it will be quite small. In this paper, we assume all the SHPPs have no reservoir, and their installed capacities are lower than 50 MW. Thus, the water-flow balance equation constraint can be formulized as below:

$$q_{in,s,t} = q_{out,s,t} = q_{s,t} + q_{spill,s,t} \tag{18}$$

where $q_{in,s}$, $q_{out,s}$ represent water inflow rate and outflow rate in scenario $s$ at time period $t$, separately (m³/s); $q_{s,t}$ denotes the volume flow ratio passing through the turbine in scenario $s$ at time period $t$ (m³/s); $q_{spill,s,t}$ represents water spillage rate (m³/s) in scenario $s$ at time period $t$.

As for SHPP's power output $P_{i,s,t}^{H}$, according to previous research [28], it can be formulated as:

$$P_{i,s,t}^{H} = \eta A q_{i,s,t} h \tag{19}$$

where $\eta$ is the hydraulic efficiency of the turbine (%); $A$ denotes the SHPP generation conversion coefficient (kg/m²s²); $h$ represents the net head of SHPP (m). Furthermore, there is no reservoir in the small hydropower station in our model, and the upstream water level cannot be controlled artificially. Therefore, the net head $h$ is approximately treated as a constant.

Also, the following water flow rate limits and power output constraints should be satisfied during the operation of SHPP at each time period.

$$q_{i,\min} \leq q_i \leq q_{i,\max} \tag{20a}$$

$$0 \le q_{spill,t} \le q_{spill,\max} \tag{20b}$$

$$p_{hi,\min} \le p_{hi,t} \le p_{hi,\max} \tag{20c}$$

Equation (20a) is the water flow rate limit which discharges from the turbine, and $q_{i,\min}$, $q_{i,\max}$ denote the lower and upper limits, respectively; Equation (20b) represents the water spillage rate limit, and $q_{spill,\max}$ is the maximum amount of water abandoned; Equation (20c) represents the *i*-th SHPP power constraint, while $P_{hi,\min}$ and $P_{hi,\max}$ denote the lower and upper limits, separately.

### 3.3.2. BESS

In term of [29], the operation cost of BESS $C_{l,t}^{BESS}$ at each time period usually denotes battery lifecycle cost, which can be modeled by a linear function as [30]:

$$C_{l,t}^{BESS} = \delta_l P_{l,t}^D \Delta t + \delta_l P_{l,t}^C \Delta t + \delta_l \eta_L E_{l,t} \tag{21}$$

where $P_{l,t}^D$, $P_{l,t}^C$ denote the discharge and charge power of BESS *l*; $E_{l,t}$ represents energy stored in BESS *l*; $\Delta t$ is the time interval factor which converts power to energy; $\eta_L$ represents the self-discharge coefficient of BESS *l*; $\delta_l$ denotes the lifecycle cost factor of BESS *l*, which can be formulized as:

$$\delta_l = \frac{I_l^{ES}}{E_{lr} \cdot (LC)_l} \tag{22}$$

where $I_l^{ES}$, $E_{lr}$ and $(LC)_l$ represent initial investment cost, rated energy capacity, life cycle number of BESS *l*, respectively.

(1)  BESS constraints

The operation of a BESS is constrained by:

$$E_{l(t+1)} = (1 - \eta_L)E_{lt} - \frac{P_{l,t}^D \Delta t}{\eta_D} + \eta_C P_{l,t}^C \Delta t, t = 0, 1, 2 \dots \tag{23a}$$

$$SOC_{l,t} = \frac{E_{l,t}}{E_{lr}} \tag{23b}$$

$$SOC_{l,\min} \le SOC_{l,t} \le SOC_{l,\max} \tag{23c}$$

$$0 \le P_{l,t}^C \le u_{l,t}^C P_{l,\max}^C \tag{23d}$$

$$0 \le P_{l,t}^D \le u_{l,t}^D P_{l,\max}^D \tag{23e}$$

$$u_{l,t}^C + u_{l,t}^D \le 1; u_{l,t}^C, u_{l,t}^D \in \{0, 1\} \tag{23f}$$

$$E_{l,0} = E_{l,end} \tag{23g}$$

where $\eta_C$, $\eta_D$ refer to charge/discharge efficiency respectively, and the default charge/discharge efficiency are constant; $SOC_{l,\min}$, $SOC_{l,\max}$, $SOC_{l,t}$ represent the lower, upper limits and the value of state-of-charge; $P_{l,\max}^C$, $P_{l,\max}^D$ refer to the maximum of charge and discharge power output; $u_{l,t}^C$, $u_{l,t}^D$ are binary variables that indicate if BESS *l* is charging ($u_{l,t}^C = 1$) or discharging ($u_{l,t}^D = 1$) during stage *t*. Equation (23a) states the energy capacity balance of BESS, which considers the net energy difference, energy losses and self-discharge loss during charging or discharging; Equations (23b)–(23d) refer to state-of-charge restrictions, and maximum limits of discharging or charging power capacities; Equation (23e) denotes that BESS cannot be charged and discharged at the same time; Equation (23f) defines the prime energy stored should be the same as the final state.

(2)    TPP constraints

The constraints of a TPP during its operation are modeled by:

$$P^T_{j,\text{min}} \le P^T_{j,t} \le P^T_{j,\text{max}} \tag{24a}$$

$$R^D_j \Delta t \le P^T_{j,t} - P^T_{j,t-1} \le R^U_j \Delta t \tag{24b}$$

where $R_j^D$, $R_j^U$ refer to the units ramping up and down limitations of TPP $j$. Equations (24a) and (24b) represent power limitation and unit ramp rate constraints of TPP $j$.

(3)    WPP constraints

$$P^W_{k,\text{min}} \le P^W_{k,t} \le P^W_{k,\text{max}} \tag{25}$$

(4)    Power balance constraints

$$\sum_{i \in \varphi_{NH}} P^H_{i,t} + \sum_{j \in \varphi_{NT}} P^T_{j,t} + \sum_{k \in \varphi_{NW}} P^W_{k,t} + \sum_{l \in \varphi_{NB}} (P^D_{l,t} - P^C_{l,t}) - \sum_{n \in \Omega_n} D_{n,t} = \sum_{n,m \in \Omega_N} b_{n,m}(\theta_{n,t} - \theta_{m,t}) \tag{26}$$

$b_{n,m}$ is the susceptance of line $n$ to $m$; $\theta_{n,t}$, $\theta_{m,t}$ represent the phase angle of bus $n$ and bus $m$ at time stage $t$; $D_{n,t}$ denotes the electricity load of bus $n$ at time period $t$.

(5)    Power transmission constraints

$$\left| b_{n,m}(\theta_{\text{n},t} - \theta_{\text{m},t}) \right| \le F^{\text{max}}_{\text{n,m}} \tag{27}$$

Equation (27) indicates transmission line power limitation; $F^{\text{max}}_{\text{n,m}}$ denotes the maximum power flow of transmission line $n$ to $m$.

(6)    Robustness constraints

Apart from the conventional robust optimization, the robustness of this model is only reflected in the constraint conditions, i.e., there are no random parameters in the objective function. Here is the set of robustness constraints.

$$
\begin{aligned}
&U := \Big\{ P_{H,W,T,D,C} = (P^H_{i,t}, P^W_{k,t}, P^T_{j,t}, P^D_{l,t}, P^C_{l,t}) : \\
&\forall \widehat{P}^W_t \in \Omega, \exists \Delta P = (\Delta P^{T+}_1, \Delta P^{T-}_1 \ldots \Delta P^{T+}_{NT}, \Delta P^{T-}_{NT}, \Delta P^D_1, \Delta P^C_1 \ldots \Delta P^D_{NB}, \Delta P^C_{NB}), \\
&\text{such that}
\end{aligned}
\tag{28a}
$$

$$\sum_{j \in \varphi_{NT}} \Delta P^T_{j,t} + \sum_{l \in \varphi_{NB}} (\Delta P^D_{l,t} - \Delta P^C_{l,t}) + \sum_{k \in \varphi_{NW}} (\widehat{P}^W_{k,t} - P^W_{k,t}) = \sum_{n,m \in \Omega_n} b_{n,m}(\overline{\theta}_{n,t} - \overline{\theta}_{m,t}),$$

$$P^T_{j,\text{min}} \le P^T_{j,t} + \Delta P^{T+}_{j,t} - \Delta P^{T-}_{j,t} \le P^T_{j,\text{max}}, \tag{28b}$$

$$R^D_j \Delta t \le P^T_{j,t} - P^T_{j,t-1} + \Delta P^{T+}_{j,t} - \Delta P^{T-}_{j,t} \le R^U_j \Delta t, \tag{28c}$$

$$0 \le P^C_{l,t} + \Delta P^C_{l,t} \le P^C_{l,\text{max}}, \tag{28d}$$

$$0 \le P^D_{l,t} + \Delta P^D_{l,t} \le P^D_{l,\text{max}}, \tag{28e}$$

$$SOC_{l,\text{min}} \le \frac{\overline{E}_{l,t}}{E_{lr}} \le SOC_{l,\text{max}}, \tag{28f}$$

$$\overline{E}_{l(t+1)} = (1 - \eta_L)\overline{E}_{lt} - \frac{(P^D_{l,t} + \Delta P^D_{l,t})\Delta t}{\eta_D} + \eta_C(P^C_{l,t} + \Delta P^C_{l,t})\Delta t, t = 0,1,2\ldots, \tag{28g}$$

$$\left| b_{n,m}(\overline{\theta}_{n,t} - \overline{\theta}_{m,t}) \right| \le F^{\text{max}}_{n,m} \Big\} \tag{28h}$$

Equation (28a) represents the power balance in re-dispatch stage, Equations (28b), (28d), (28e), and (28f) indicate the power limit of TPPs and BESSs in re-dispatch, respectively. Notice that Equation (28c) denotes the ramp constraints of TPPs. Equation (28g) denotes BESS energy stored change, which contains net energy injection and losses in re-dispatch. And Equation (28h) shows the transmission limitation in real-time stage.

### 3.4. Model Reformulation

However, the problem (Equations (15)–(28)) described above cannot be solved directly. Therefore, in terms of the previous research results [31,32], we should use reformulation to make the problem easy to deal with. The problem (Equations (15)–(28)) can be equivalently stated as follows:

$$(\text{MP}) : \min f = \sum_{t=1}^{T} \left( \sum_{i=1}^{NH} \sum_{s=1}^{C} \rho_s C_i P_{i,t}^H \Delta t + \sum_{j=1}^{NT} C_j P_{j,t}^T \Delta t + \sum_{k=1}^{NW} C_k P_{k,t}^W \Delta t + \sum_{l}^{NB} C_{l,t}^{BESS} \right) \tag{29}$$

s.t. Constraints (18)–(27).

$$P_{j,\min}^T \leq P_{j,t}^T + \Delta P_{j,v,t}^T \leq P_{j,\max}^T, \forall v \in V \tag{30a}$$

$$R_j^D \Delta t \leq P_{j,t}^T - P_{j,t-1}^T + \Delta P_{j,v,t}^T \leq R_j^U \Delta t, \forall v \in V \tag{30b}$$

$$0 \leq P_{l,t}^C + \Delta P_{l,v,t}^C \leq P_{l,\max}^C, \forall v \in V \tag{30c}$$

$$0 \leq P_{l,t}^D + \Delta P_{l,v,t}^D \leq P_{l,\max}^D, \forall v \in V \tag{30d}$$

$$(\text{SP}) : \psi = \max_{\widehat{P}_t^W \in \Omega} \min_{\Delta P_{j,t}^{T+}, \Delta P_{j,t}^{T-}, \Delta P_{l,t}^D, \Delta P_{l,t}^C, \overline{E}_t, \overline{\theta}_{n,t}} \sum_{t=1}^{T} \left( \sum_{j=1}^{NT} a_j (\Delta P_{j,t}^{T+} - \Delta P_{j,t}^{T-}) + \sum_{l=1}^{NB} b_j (\Delta P_{l,t}^D - \Delta P_{l,t}^C) \right) \text{ s.t.}$$

Constraints (14) and (28).

As can be seen from the reformulation, the problem mentioned above is divided into two stages, which includes a master problem (MP) and a sub-problem (SP). Where $V$ is the index set for worst uncertainty scenario $\widehat{P}_t^W$ which are dynamically gained from the process of handling SP. $a_j$, $b_j$ represent the adjustment cost coefficient of TPPs and BESSs for accommodating uncertainty, respectively. The inner level objective function of SP refers to find the optimal power modulation of flexible generators (TPPs, BESSs) in real-time stage under the worst uncertainty case, and the outer level denotes that seeking for the worst wind power actual output under the fixed budget parameter $\Gamma$. In such a context, the original problem is transformed into two-stage robust optimization problem.

### 3.5. Solution Methodology

As for the solution of the two-stage robust optimization problem, several researches have been reported in the previous literature [33–35]. In [33], a constraint and column generation (C&CG) method is presented and comprehensively analyzed. This technique is a mature and fast method to deal with the robust optimization. Therefore, the detailed solutions procedures of CCG algorithm are not displayed. The application of this algorithm in this paper is formally proposed as follows:

(1)　$V \leftarrow \varnothing, v \leftarrow 1, \Psi \leftarrow +\infty$, define feasibility tolerance;

(2)　While $\psi \geq \Delta\varepsilon$ do;

(3)　Solve (MP), and obtain optimal where $V$ is the index set for worst uncertainty points $\widehat{P}_t^W$ which are dynamically gained from the process of handling SP;

(4)　Solve (SP) with $P_{i,t}^H, P_{k,t}^W, P_{j,t}^T, P_{l,t}^D, P_{l,t}^C, \forall i, k, j, l, t$, find solution $(\psi, \widehat{P}_{k,t}^W, \Delta P_{j,t}^{T+}, \Delta P_{j,t}^{T-} \Delta P_{l,t}^D, \Delta P_{l,t}^C)$;

(5)　$V \leftarrow V \cup v, v \leftarrow v+1$;

(6)    End while.

The solution procedure of overall two-stage stochastic robust optimization problem is shown in Figure 3.

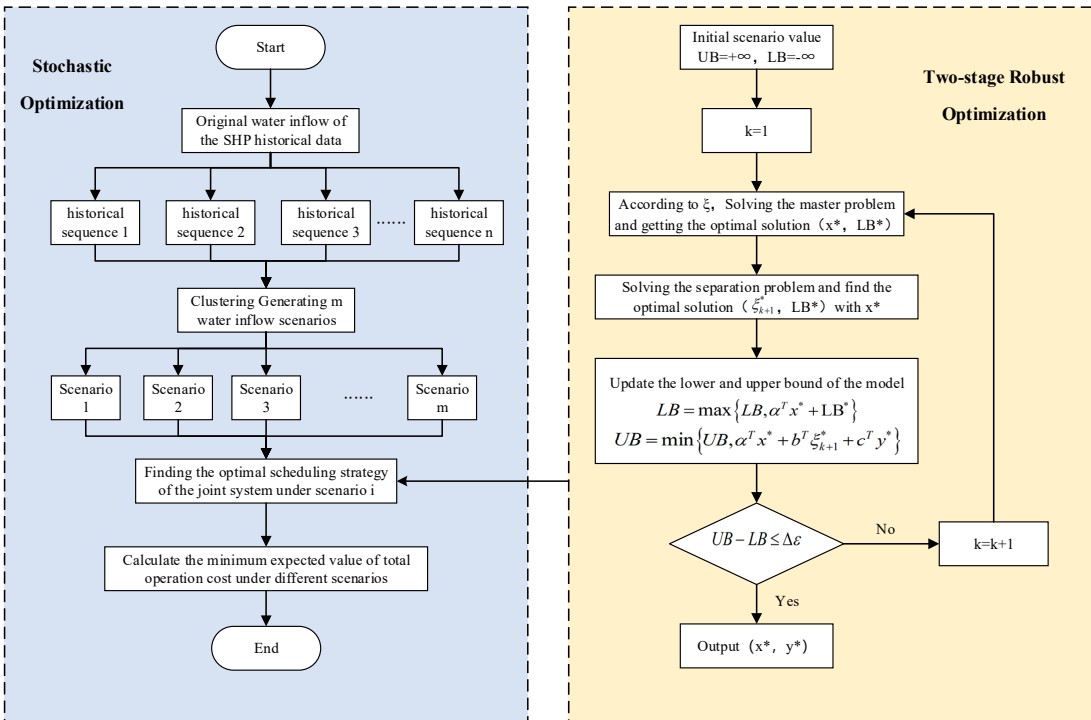

**Figure 3.** The solution procedure of two-stage stochastic robust optimization problem.

## 4. Numerical Example

A numerical case is simulated to verify the applicability of the model in Figure 3. Firstly, the setting of the general basic background should be completed. In terms of whether the uncertainty of wind power output is considered, it is divided into two different cases to form two problems, namely, deterministic optimization and uncertain optimization. Deterministic optimization is a general linear integer programming problem, which can be solved directly. Uncertainty optimization is solved by a two-stage stochastic robust optimization as shown in Figure 3. The solution idea can be roughly divided into two parts. In the first part, the FCM-CCQ method is used for clustering analysis of the inflow scenario of small hydropower (corresponding to the process in the left half of Figure 3). In this numerical example, 100 groups of real historical water inflow data of small hydropower are input, and the final clustering result is determined by FCM-CCQ algorithm. In the second part, based on each type of incoming water scenario, the two-stage robust optimization method (corresponding to the process in the right half of Figure 3), is used to identify the scenario with the greatest negative impact on the power system according to the uncertainty interval of the wind power output. On this basis, the optimal scheduling strategy is obtained. This strategy can be understood as, that no matter how the actual output of wind power changes in the future, the total cost of system operation will not exceed the total cost under the optimal strategy. In the end, the system operation cost calculated under the optimal strategy under each kind of small hydropower inflow scenario is multiplied by the corresponding occurrence probability of this category, and then the expected operation cost of the system is obtained by summing them.

### 4.1. Scenario Set

To analyze the practicability of the proposed algorithm in solving the uncertainty problems of SHPPs and WPPs, this paper sets three simulation scenarios.

### 4.1.1. Case 1

Only in view of the uncertainty of water inflow, we propose k-means method to cluster the scenario of incoming water from small hydropower, and reduce the total number to 20.

Based on 20 kinds of inflow scenarios, economic scheduling is carried out under the condition that the wind power output is all of a certain value. The scenario mainly analyzes the influence of different incoming water quantity on system operation cost in a day-ahead market.

### 4.1.2. Case 2

Comprehensive scenario, not only considering the uncertainty of water inflow, but also wind power output. In the case of uncertain wind power output, the two-stage robust optimization algorithm is adopted to calculate the expected minimum total operation cost of 20 scenarios. This case involves the scheduling optimization of unbalanced power in the real-time market, which is more complicated than the previous two cases. Furthermore, compare the scheduling results with a different budget parameter Γ.

### *4.2. Power System Parameter Setting*

In this section, we present an artificial simulation to test the proposed model using Matlab R2015b and CPLEX Optimization Studio v12.8. An IEEE 30-bus test system with three SHPPs, three WPPs, two TPPs and two BESSs is adopted as an example. The detailed data refers to the website in [36].

The network topology of the IEEE 30-bus test system in this paper is shown in Figure 4. For the sake of simplicity, it is assumed that the capacity limitation of power flow for all transmission lines are 35 MW. Besides, the physical and economic parameters related to distributed generators are shown in Table 1. The operating parameters of TPPs and BESSs are tabulated in Tables 2 and 3. These relevant parameters are collected from [12,28,29]. The predicted load curve for the overall power system for 24 h are drawn in Figure 5. Detailed load data are shown in Tables A1–A3 in Appendix A. This manuscript assumed that load in any bus is inelastic, without considering shedding load. As seen from Figure 5, there are multiple spikes in the whole load curve, including the 4th, 11th, 13th, 15th and 22nd time periods. These peaks will have an important impact on the overall scheduling process, and our subsequent experimental results will be analyzed in combination with these load characteristics. Besides, Figure 6 denotes the forecasted power output of the three WPPs.

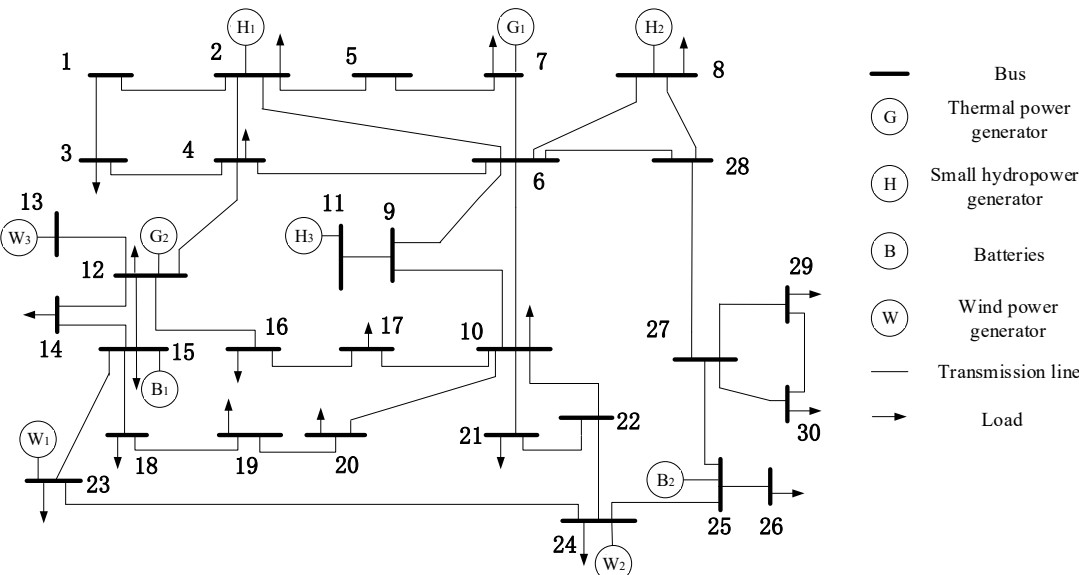

**Figure 4.** The schematic structure of the Institute of Electrical and Electronics Engineers (IEEE) 30-bus system.

**Table 1.** Distributed generators parameters.

| Distributed Generators | Pmin (MW) | Pmax (MW) | Iph,Ipw (¥/MWh) | Opw,Oph (¥/MWh) | Output Conversion Coefficient | Net Head (m) | qmin (m³/s) | qmax (m³/s) |
|---|---|---|---|---|---|---|---|---|
| **SHPP1** | **0** | **20** | **922.35** | **60** | **8.5** | **60** | **0** | **39.2** |
| SHPP2 | 0 | 15 | 1383.52 | 90 | 8.58 | 54.5 | 0 | 32 |
| SHPP3 | 0 | 12 | 1460.38 | 95 | 8.2 | 48 | 0 | 30.4 |
| WPP1 | 0 | 15 | 1537.25 | 100 | — | — | — | — |
| WPP2 | 0 | 13 | 2305.87 | 150 | — | — | — | — |
| WPP3 | 0 | 10 | 2690.18 | 175 | — | — | — | — |

**Table 2.** Thermal power generators parameters.

| Thermal Power Generators | Cj (¥/MWh) | PTmin (MW) | PTmax (MW) | Rampupmax | Rampdownmax |
|---|---|---|---|---|---|
| TPP1 | 550 | 5 | 50 | 20 | 20 |
| TPP2 | 600 | 3 | 45 | 15 | 15 |

**Table 3.** Energy storing device parameters.

| Parameter | BESS1 | BESS2 |
|---|---|---|
| Initial energy storage (MWh) | 5 | 5 |
| Rated energy capacity (MWh) | 20 | 20 |
| Discharge/charge efficiency | 0.95 | 0.95 |
| PCmax (MW) | 15 | 10 |
| PDmax (MW) | 15 | 10 |
| Self-discharge rate | 0.05 | 0.0.5 |
| LC (times) | 100 | 150 |
| IC (¥) | 30 | 50 |

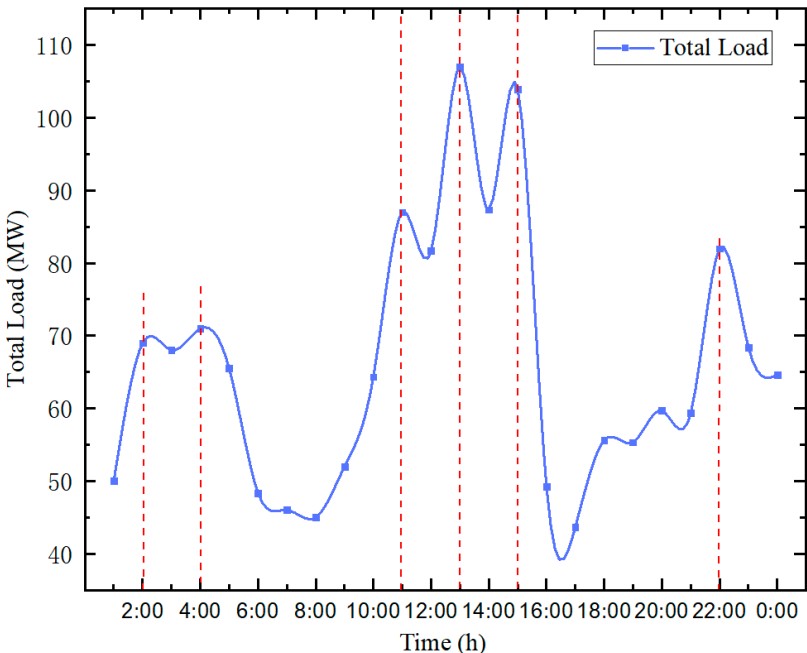

**Figure 5.** Total forecasted load for the overall power system in day-ahead.

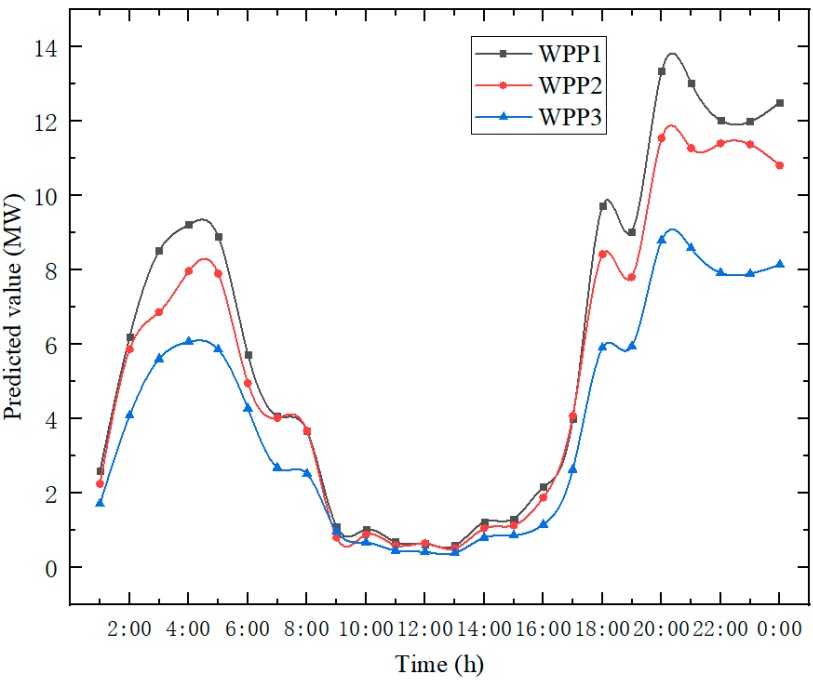

**Figure 6.** The prediction of three WPPs' output.

### 4.3. Incoming Water Scenario Clustering

Before day-ahead scheduling optimization, FCM-CCQ is utilized to cluster the incoming water scenarios of small hydropower firstly. Specifically, according to the actual collection of 100 groups of daily incoming water quantity historical data, using the FCM method to cluster the data with the final clustering number of 1–30, successively. In terms of the different number of categories, 30 classification results are obtained at last. Then the CCQ algorithm is used to evaluate and score each classification result, and the correlation curve of clustering number and average comprehensive quality value is drawn, as shown in Figure 7. From the figure, we can easily find the inflection point of the curve. The number of clustering scenarios corresponding to this point is 20, that is to say, clustering 100 groups of incoming water scenarios into 20 is the best clustering scheme. Figure 8 shows water inflow of three SHPPs under 20 typical scenarios.

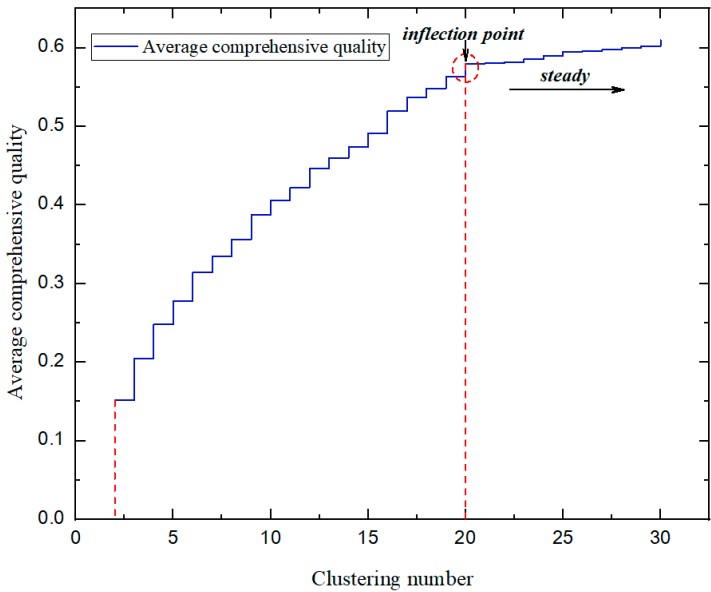

**Figure 7.** The correlation curve of clustering number and average comprehensive quality value.

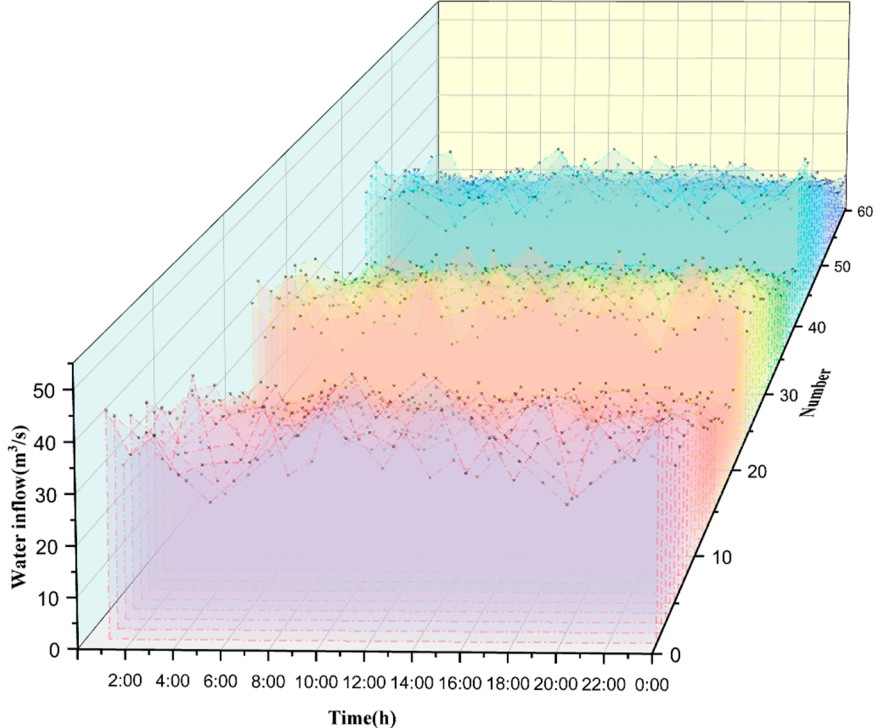

**Figure 8.** Water inflow of 20 scenarios (SHPP1: No. 1–20, SHPP2: No. 21–40, SHPP3: No. 41–60).

Generally speaking, the ordinary clustering analysis methods such as K-means and the FCM, will be given in advance to aggregate number of categories. For example, the 100 scenario data is clustering to 20 categories, of course, it can use the general clustering method, but cannot explain why it is divided into 20 categories. Is this clustering scheme the best one? These questions cannot be answered by the ordinary clustering method. However, CCQ algorithm which is adopted to evaluate each classification result can solve this issue and select the best scheme via indicators calculation. Therefore, the conventional clustering method combined with the CCQ algorithm is better than only using the general clustering method.

*4.4. Results and Discussion*

After the pre-clearing of the day-ahead market without considering the uncertainty influence on the power system in the first stage, Figure 9a,b show the results of power output of each power plant and BESS separately over the whole scheduling horizon, based on the predictive value of WPP available power in one of the flood period scenarios. In Figure 9b, BESSs with positive output indicate that they are in the process of discharge. On the contrary, they are in charge.

From Table 1, we can get that the operation cost of SHPPs and WPPs is lower than TPPs. With this in mind, SHPPs and WPPs should be given priority to generation, which can be demonstrated in Figure 9a. However, due to the limitation of incoming water and wind, their power output presents a certain volatility. When the total available power of SHPPs and WPPs is not enough to meet the total system load, BESSs will discharge to fill the power shortage at first, compared with TPPs (e.g., BESSs discharge automatically at the 4th, 5th, 13th, 14th and 15th time stages). When the BESS energy stored also reaches the lower limit of its rated capacity, that is to say, the BESS cannot continue to discharge anymore, the TPPs will begin to generate to fill the power gap. This is the reason why TPP1 outputs appear as the peak values between 11:00 and 15:00.

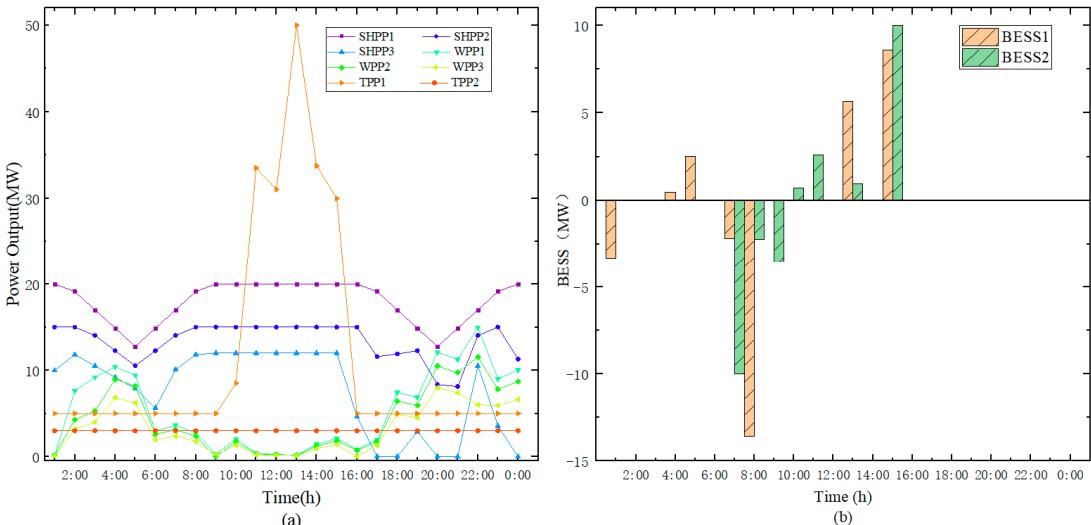

**Figure 9.** The results over the whole scheduling horizon in first stage: (**a**) The scheduling output of power generation components except BESSs in the first phase; (**b**) The scheduling output of BESSs in the first phase.

In the second re-dispatch stage, considering the uncertainty of wind power output, power generators with some flexibility, such as TPPs and BESSs, can adjust the power generation according to the power deviation value of wind power, and maintain the real-time balance between supply and demand. In the meantime, as the size of the uncertain set is fixed, the worst case of the wind power output that maximizes the total system operation cost is obtained. This worst case means that the ISO considers the worst wind power output in a certain confidence interval. When scheduling under the worst case, the total operational cost is not higher than the operating cost at this time, no matter how the actual wind power output changes.

Figure 10a–c illustrate the confidence intervals for the power output of WPP1, WPP2 and WPP3 in the case of, respectively, where the upper and lower curves correspond to the predicted upper and lower limit of the wind power output separately. The detailed data of the upper and lower limits of the output range are shown in Table A4 of Appendix A. Also, this interval is the boundary of the uncertainty set which was formulated as Equation (14). Notably, the shaded portion is the confidence interval for the predicted wind power output within 24 h. The red dotted line in the middle indicates the worst case power output, and the actual maximum power output curve of each wind farm as well. Considering this worst case of the WPP, the upper limit of the WPP output power can be updated to the red line, then start optimizing the day-ahead scheduling again.

Repeat the above process until the total running cost no longer rises. The scheduling result is shown in Figure 11. Compared with Figure 9, the output power of WPPs are reduced. SHPP2, TPPs, and BESSs are adjusted accordingly to adapt to wind power during the period when wind power pre-output is reduced. Obviously, during peak load hours, TPP1, BESS1, and BESS2 both increase output power. Because of the high thermal power generation and energy storage costs compared to renewable energy generation in our setup, the total operating cost is significantly improved.

However, in spite of the fact that thermal power and energy storage systems have some capacity occupied in the day-ahead market, the remaining spare capacity is still sufficient to balance the deviation of wind power output power fluctuations in the re-dispatch stage. In Figure 12, the solid line indicates the total available power that is initially scheduled by the ISO, in accordance with the predicted value of the WPP output power, in order to satisfy the load demand of each bus. In the real-time phase, the wind power output deviates from the predicted value, so that the actual total output power curve (dashed line) changes. The deviation is the orange part in Figure 12. It can be concluded that the deviation value is generated at the peak load. At this point, if the WPPs' output has

a decrease, it is necessary to call the standby unit to compensate for the deviation. In fact, during the off-peak period, some of the power generation capacity of SHPP3 is not fully utilized, but in order to drive the total cost to a higher direction, the deviation purely occurs during the peak period. In this sense, it well explains the confusion that the deviation value only appears during the peak load period.

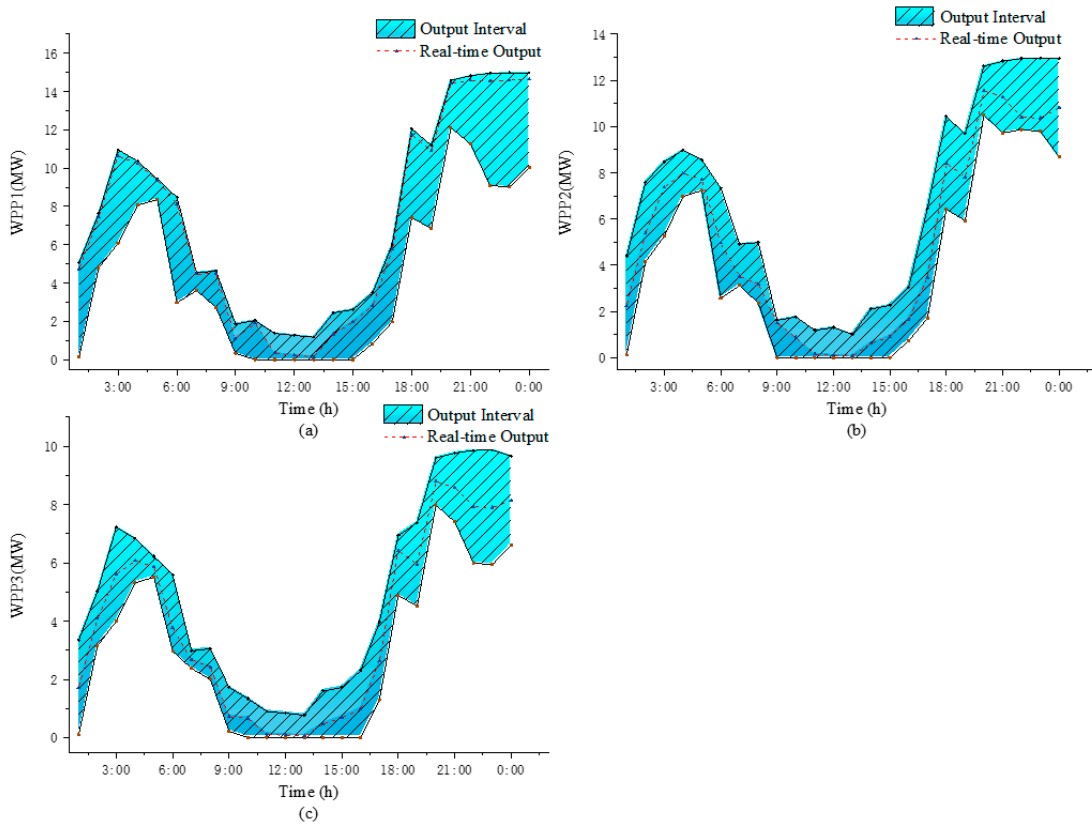

**Figure 10.** The confidence intervals for the power output of WPPs ($\Gamma = 1$): (**a**) The confidence intervals for the power output of WPP1; (**b**) The confidence intervals for the power output of WPP2; (**c**) The confidence intervals for the power output of WPP3.

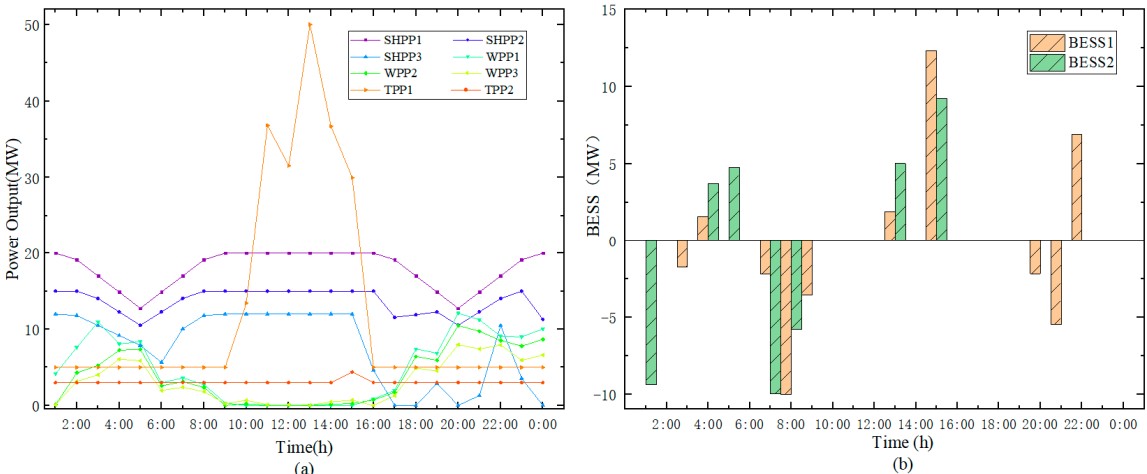

**Figure 11.** The results over the whole scheduling horizon in second stage: (**a**) The scheduling output of power generation components except BESSs in the second stage; (**b**) The scheduling output of BESSs in the second stage.

Contrapose the deviation generated in the real-time stage, Figure 13 shows the scheduling strategy in detail for the ISO to deal with the problem in the overall re-scheduling horizon. "BESS1_PD_A"

represents the adjustment when BESS1 is discharged. Furthermore, a positive value indicates the increased discharge at that moment, and a negative value indicates the decrease. Similar to the BESS1, "TPP1_A" denotes the variation of TPP1 power output, compared with the day-ahead dispatch in Figure 9a. "TPP1_OS" and "TPP1_AS" illustrate the day-ahead scheduling curve and the real-time scheduling curve of TPP1. It can be observed that the changes between the two curves are exactly the adjustment of TPP1 (e.g., at 15:00 and 22:00). According to Equations (23a) and (23b), we know that both BESS1 and BESS2 are constrained by their energy stored capacity. Furthermore, the most capacity of BESSs have already been dispatched in the day-ahead clearing process. As shown in Figure 14, BESS1 is charged to its rated capacity at time period nine, and then continuously discharged from ten to fifteen. Because of BESS1 discharging in advance, this results in a lack of the energy stored to discharge anymore. This is why the adjustment of BESS1 appears as a negative value at the 15th time stage. At this moment, this power shortage is filled by TPP1.

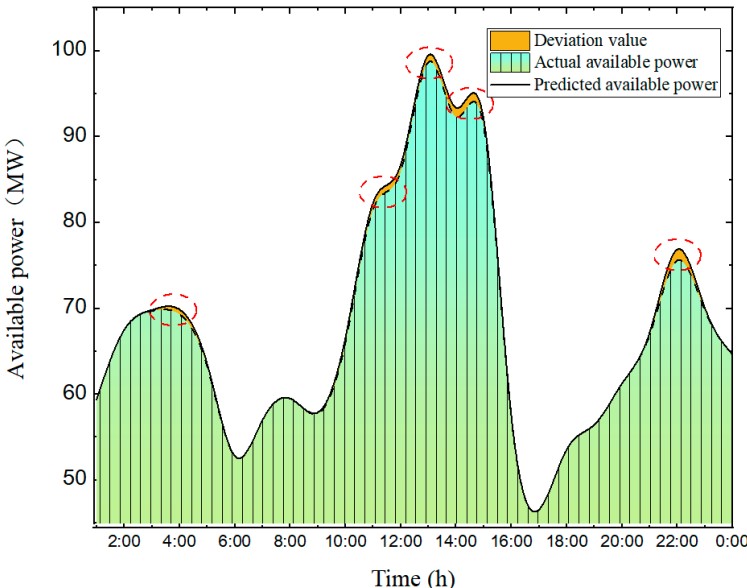

**Figure 12.** The deviation between actual available power and predicted available power.

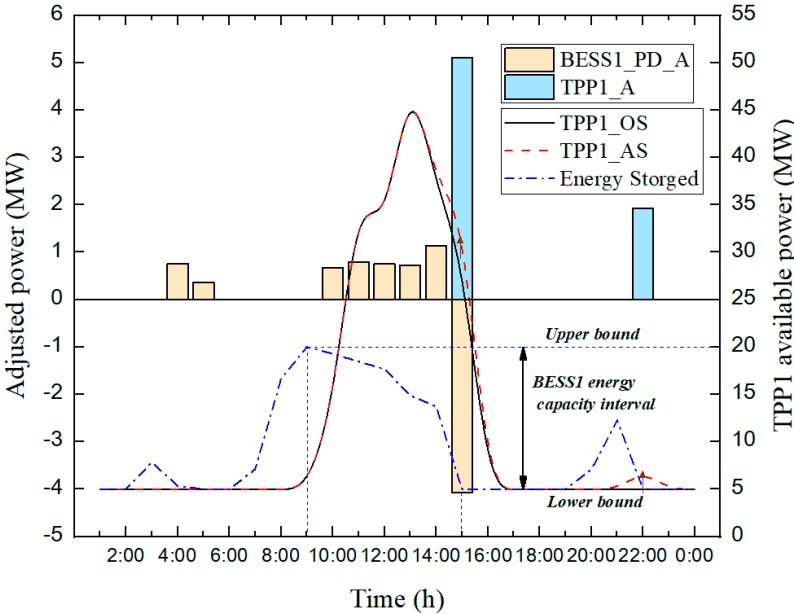

**Figure 13.** The scheduling strategy for the independent system operator (ISO) in the second stage.

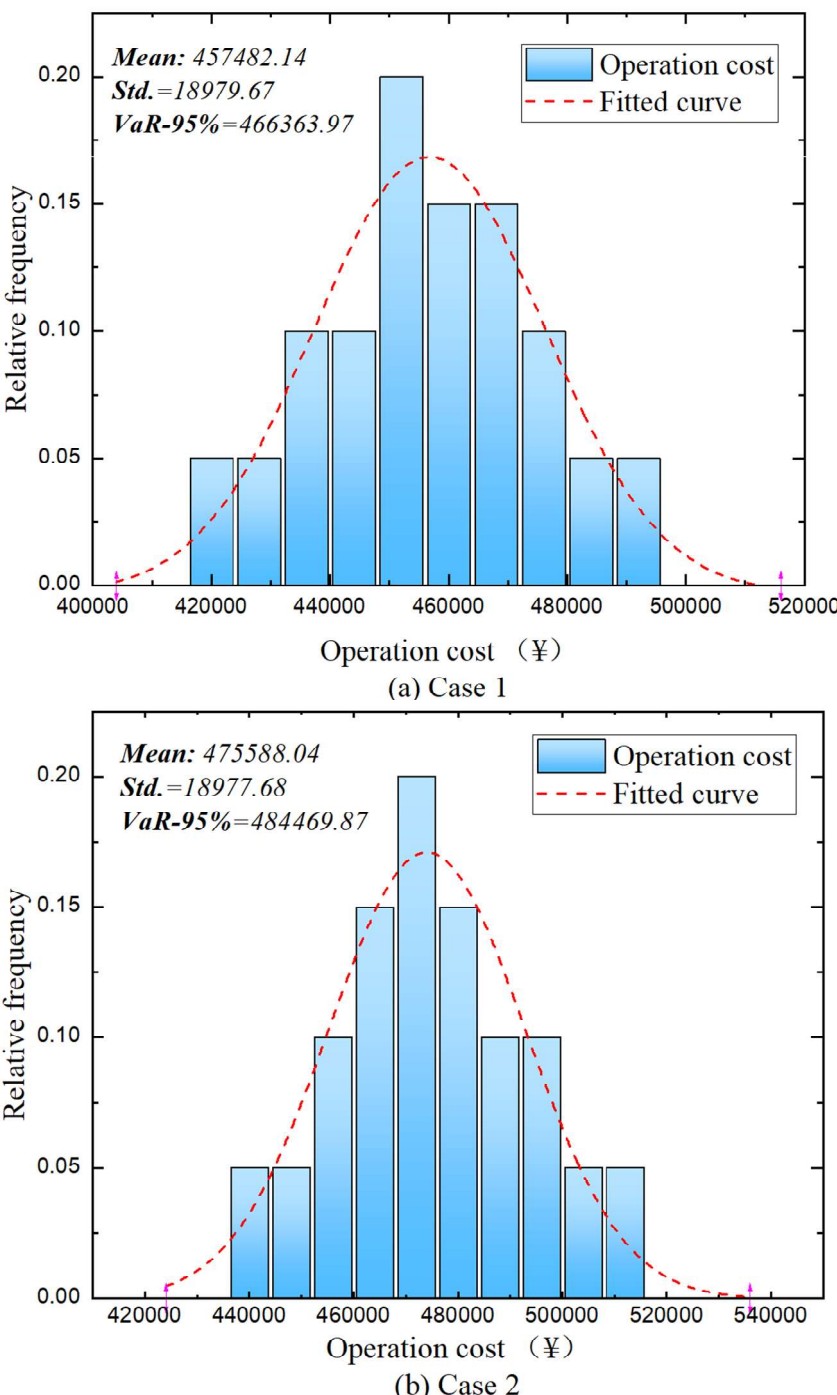

**Figure 14.** Distributions of system operation cost in day-ahead market ($\Gamma = 1$): (**a**) Distributions of system operation cost in day-ahead market in Case 1; (**b**) Distributions of system operation cost in day-ahead market in Case 2.

Figure 14 shows the system operation cost corresponding to whether the wind power uncertainty is considered under different inflow scenarios. Specifically, in case 1, it is assumed that the wind power output is determined, and the system costs of different inflow scenarios are calculated as shown in Figure 14a. If wind power uncertainty is considered in case 2, the trial operation cost of solving different incoming water scenarios using two-stage robust optimization is shown in Figure 14b. The cost in both cases does not include the cost of real-time market rescheduling, but only the system operating cost pre-cleared in day-ahead. Specially, due to the possible scheduling process in the real-time market

stage, the flexible unit reserves spare capacity to cope with the instability of the wind power output in case 2. The total trial run cost of each scenario can be seen as generally obeying the normal distribution. The expected operating costs of the two cases were ¥457,482.14 and ¥4,775,588.04, with standard deviations of ¥18,979.67 and ¥18,977.68, respectively.

In addition, we adopt the crucial indicator—Value-at-risk (VaR) to estimate the risk of different operation strategies associated with diverse water inflow scenarios at the 95% confidence level. The VaR-95% means that the operating cost will not be higher than the expected value at a 95% confidence rate, which are ¥466,363.97 and ¥484,469.87 in case 1 and case 2 respectively.

It can be seen from the above index data that case 2 has an operating cost increase of 3.96% ((4,775,588.04−457,482.14)/457,482.14) and a VaR-95% increase of 3.88% ((484,469.87−466,363.97)/466,363.97) over case 1, which means that once the uncertainty of wind power output is considered, the expected operating cost of the system will increase. The reason for this result is that when the scheduling problem is solved by robust optimization, ISO must take into account the worst case of wind power output before day-ahead scheduling. Furthermore, the power system should actively reserve enough spinning reserve capacity to adjust the deviation power in the real-time phase, which undoubtedly increases the total operating cost. In other words, the conservative scheduling strategy will inevitably lead to the increase of the total system operating cost, but it will greatly enhance the security and reliability of the power system.

### 4.5. Sensibility Analysis

In this section, we take into account the influence of the difference of the budget parameter $\Gamma$ on the system operational cost. Therefore, the sensitivity analysis of this factor is carried out here. First of all, let us discuss the role of this factor. The value of $\Gamma$ determines the size of the wind power output uncertainty set. The smaller $\Gamma$ is, the smaller the size of uncertainty set. To a certain extent, $\Gamma$ is also a key indicator of ISO risk assessment for the uncertainty of wind power. When the value of $\Gamma$ is larger, the uncertainty of the wind power output is estimated by ISO. At this time, ISO supposes that the uncertainty will cause some impact on the operation of the system. Therefore, in the clearing stage, the spare reserve capacity for the rescheduling phase is used to maintain the balance and stability of the power supply and demand. A more conservative production scheduling strategy will be adopted, which will directly lead to an increase in system operating costs, but the security of the system operation can be also guaranteed. Especially, when $\Gamma = 0$, it means that the forecasted value of each WPP power output is a determined value that is equivalent to the middle value of its power output confidence interval, and the day-ahead clearing model converts to a deterministic ED problem.

However, the uncertainty problem exists objectively in a power system that has a penetration of RESs. Without enough consideration of the uncertainty, ISO might find it hard to accommodate the deviations caused by WPPs in real-time scheduling. Thus, it is vital for the day-ahead clearing process to compare with final operation results under different $\Gamma$ values. Table 2 shows the operation results associated with different $\Gamma$ values.

As shown in Table 4, we can observe the total expected system operation cost increases with the increase of budget parameter $\Gamma$. When $\Gamma = 1$, the optimal operating cost of the system is higher than the optimal value of $\Gamma = 0$. This is because the total operating cost obtained by robust optimization is conservative when considering the uncertainty. Although the cost at $\Gamma = 0$ is very low, ignoring the uncertainty of wind power will cause great risks to the entire power system. If the actual output of the wind turbine the next day is significantly different from the predicted value of wind power, it will cause greater damage to the power system. This shows that considering the uncertainty in the optimization process can reduce the system operation risk, but it will have a corresponding impact on the total cost. However, when considering the uncertainty, as the robustness factor $\Gamma$ increases, the total cost of the system will increase with the robust optimization. Therefore, it is the best choice for ISO to select favorable scheduling decisions for power systems according to the actual situation. In addition, the third column means that whether the uncertainty can be accommodated in the real-time stage

when the day-ahead optimal scheduling is obtained. Comparing the cases of $\Gamma = 2$ and $\Gamma = 3$, we can find that the expected operation cost in both cases is same, which means there is no uncertainty that cannot be accommodated.

**Table 4.** Operation results under different $\Gamma$ values.

| Budget Parameter | Total Expected System Operation Cost | Uncertainty Set That Cannot Be Accommodated |
|---|---|---|
| $\Gamma = 0$ | 468,824.84 | Yes |
| $\Gamma = 0.5$ | 474,383.23 | Yes |
| $\Gamma = 1$ | 475,588.04 | Yes |
| $\Gamma = 2$ | 476,710.37 | No |
| $\Gamma = 3$ | 476,710.37 | No |

## 5. Conclusions

In this paper, a two-stage stochastic robust optimization approach for day-ahead market operation dispatching of the distributed power system, with high renewable penetration that consists of SHPPs, WPPs, BESSs, and TPPs, is proposed. Before optimization, the FCM-CCQ clustering evaluation method is used to cluster the historical water quantity data of each SHPP into the optimal scenario number at first. Based on these typical inflow scenarios, the first-stage (pre-clearing process) is implemented with the control objective of minimizing the expected operation cost of the power system, and the operation is scheduled when the wind power output is assumed to be deterministic, that is, the main problem of optimization. In the second stage (re-dispatch stage), namely the subproblems, we should find the worst case of realizing wind power uncertainty that damages the system stability most from the wind power uncertainty set. After that, the worst wind power scenario is substituted into the main problem to obtain the total expected operating cost of the system simultaneously, which is the optimal result, considering the fluctuation of small hydropower and wind power output. Then, a numerical example is utilized to verify the proposed model's practicability. Obviously, simulation results show that the optimal expected value obtained by the two-stage stochastic robust optimization model is not less than the total expected cost of the system in any case. Besides, we can find that the fluctuation of wind power output has a huge impact on the power system. Through robust optimization, ISO can adequately consider the interference of uncertain factors, and make use of the flexibility of thermal power and energy storage facilities to generate electricity to accommodate the deviation caused by wind power in the second stage, so as to effectively reduce its economic risk. Compared with previous studies, the research in this paper is advanced in: (1) The collaborative operation optimization model of a distributed power system containing hydro-wind hybrid distributed energy and energy storage facilities is constructed; (2) a novel scene clustering method (FCM-CCQ) is proposed, which can not only cluster, but also evaluate whether the scene clustering number is the best one; (3) the uncertainty/intermittency of renewable energy is fully considered, and the two-stage stochastic robust optimization method is adopted to significantly reduce the impact of inaccurate renewable energy prediction on the power system and ensure the safe and stable operation of the system.

Furthermore, the proposed scheduling model adopts a conservative scheduling strategy to avoid the adverse impact of intermittency and randomness of renewable energy generation on the power system to some extent, and at the same time promotes the absorption of distributed renewable energy. This shall pave the way for large-scale absorption of renewable energy and sustainable development of the society in the future.

Finally, the model established in this paper is relatively simple and ideal, but it is hoped that the work can provide useful reference for renewable energy consumption and the day-ahead dispatching operation of the distributed power system. In addition, further research can be carried out in the direction of more complex distributed systems with the real-time change of power load demand and the response of customer demands in the future.

**Author Contributions:** J.D. supervised the proposed research work, whereas P.Y. conceptualized the proposed research work, collected the data, programmed and analyzed the methodology, and written the original draft. S.N. helped in review writing and editing with validation of research work.

**Funding:** This research was funded by Beijing social science foundation research base project (18JDGLB037).

**Conflicts of Interest:** The authors declare no conflict of interest.

## Appendix A

**Table A1.** Hourly load data for each bus of the IEEE test system (a).

| Time | Bus 1 | Bus 2 | Bus 3 | Bus 4 | Bus 5 | Bus 6 | Bus 7 | Bus 8 | Bus 9 | Bus 10 |
|------|-------|-------|-------|-------|-------|-------|-------|-------|-------|--------|
| 1:00 | 0 | 4.616183 | 1.279391 | 2.178423 | 0 | 0 | 4.806362 | 6.051176 | 0 | 1.86722 |
| 2:00 | 0 | 4.339212 | 1.202628 | 2.047718 | 0 | 0 | 6.517981 | 7.688105 | 0 | 3.755187 |
| 3:00 | 0 | 4.246888 | 1.17704 | 2.004149 | 0 | 0 | 6.421853 | 7.567082 | 0 | 3.717842 |
| 4:00 | 0 | 4.523859 | 1.253804 | 2.134855 | 0 | 0 | 6.710235 | 7.930152 | 0 | 3.829876 |
| 5:00 | 0 | 4.570021 | 1.266598 | 2.156639 | 0 | 0 | 5.558299 | 7.590664 | 0 | 3.648548 |
| 6:00 | 0 | 4.459232 | 1.235892 | 2.104357 | 0 | 0 | 4.642946 | 5.845436 | 0 | 1.803734 |
| 7:00 | 0 | 4.246888 | 1.17704 | 2.004149 | 0 | 0 | 4.421853 | 5.567082 | 0 | 1.717842 |
| 8:00 | 0 | 4.154564 | 1.151452 | 1.960581 | 0 | 0 | 4.325726 | 5.446058 | 0 | 1.680498 |
| 9:00 | 0 | 4.80083 | 1.330567 | 2.26556 | 0 | 0 | 4.998617 | 6.293223 | 0 | 1.941909 |
| 10:00 | 0 | 5.939488 | 1.64615 | 2.802905 | 0 | 0 | 6.184186 | 7.785846 | 0 | 2.40249 |
| 11:00 | 0 | 8.032158 | 2.226141 | 3.790456 | 0 | 0 | 8.363071 | 10.52905 | 0 | 3.248963 |
| 12:00 | 0 | 7.539765 | 2.089673 | 3.558091 | 0 | 0 | 7.850392 | 9.883587 | 0 | 3.049793 |
| 13:00 | 0 | 9.878631 | 2.737898 | 4.661826 | 0 | 0 | 10.28562 | 12.94952 | 0 | 3.995851 |
| 14:00 | 0 | 8.062932 | 2.23467 | 3.804979 | 0 | 0 | 8.395113 | 10.56939 | 0 | 3.261411 |
| 15:00 | 0 | 9.592427 | 2.658575 | 4.526763 | 0 | 0 | 9.987621 | 12.57434 | 0 | 3.880083 |
| 16:00 | 0 | 4.542324 | 1.258921 | 2.143568 | 0 | 0 | 4.729461 | 5.954357 | 0 | 1.837344 |
| 17:00 | 0 | 4.031466 | 1.117335 | 1.90249 | 0 | 0 | 4.197556 | 5.284693 | 0 | 1.630705 |
| 18:00 | 0 | 3.936065 | 1.090894 | 1.857469 | 0 | 0 | 7.098225 | 8.159636 | 0 | 3.592116 |
| 19:00 | 0 | 3.908368 | 1.083218 | 1.844398 | 0 | 0 | 7.069387 | 7.123329 | 0 | 4.580913 |
| 20:00 | 0 | 4.216113 | 1.168511 | 1.989627 | 0 | 0 | 7.389811 | 9.52674 | 0 | 3.705394 |
| 21:00 | 0 | 4.003769 | 1.109659 | 1.889419 | 0 | 0 | 7.168718 | 9.248386 | 0 | 4.619502 |
| 22:00 | 0 | 7.570539 | 2.098202 | 3.572614 | 0 | 0 | 7.882434 | 9.923928 | 0 | 3.062241 |
| 23:00 | 0 | 6.893154 | 1.356155 | 2.309129 | 0 | 0 | 7.094744 | 8.414246 | 0 | 3.979253 |
| 0:00 | 0 | 5.967185 | 1.653827 | 2.815975 | 0 | 0 | 6.213024 | 7.822153 | 0 | 2.413693 |

**Table A2.** Hourly load data for each bus of the IEEE test system (b).

| Time | Bus 11 | Bus 12 | Bus 13 | Bus 14 | Bus 15 | Bus 16 | Bus 17 | Bus 18 | Bus 19 | Bus 20 |
|------|--------|--------|--------|--------|--------|--------|--------|--------|--------|--------|
| 1:00 | 0 | 2.80083 | 0 | 1.936376 | 2.282158 | 1.469571 | 2.42047 | 1.417704 | 2.506916 | 1.244813 |
| 2:00 | 0 | 2.63278 | 0 | 3.820194 | 4.145228 | 3.381397 | 4.275242 | 3.332642 | 4.356501 | 3.170124 |
| 3:00 | 0 | 2.576763 | 0 | 3.781466 | 4.099585 | 3.352006 | 4.226833 | 3.304288 | 4.306362 | 3.145228 |
| 4:00 | 0 | 2.744813 | 0 | 3.897649 | 4.236515 | 3.44018 | 4.372061 | 3.38935 | 4.456777 | 3.219917 |
| 5:00 | 0 | 3.572822 | 0 | 2.717012 | 3.059336 | 2.254876 | 3.796266 | 2.203527 | 3.281846 | 2.732365 |
| 6:00 | 0 | 2.705602 | 0 | 1.870539 | 2.204564 | 1.419606 | 2.338174 | 1.369502 | 2.42168 | 1.20249 |
| 7:00 | 0 | 2.576763 | 0 | 1.781466 | 2.099585 | 1.352006 | 2.226833 | 1.304288 | 2.306362 | 1.145228 |
| 8:00 | 0 | 2.520747 | 0 | 1.742739 | 2.053942 | 1.322614 | 2.178423 | 1.275934 | 2.256224 | 1.120332 |
| 9:00 | 0 | 2.912863 | 0 | 2.013831 | 2.373444 | 1.528354 | 2.517289 | 1.474412 | 2.607192 | 1.294606 |
| 10:00 | 0 | 3.603734 | 0 | 2.491471 | 2.936376 | 1.890848 | 3.114338 | 1.824112 | 3.225565 | 1.60166 |
| 11:00 | 0 | 4.873444 | 0 | 3.369295 | 3.970954 | 2.557054 | 4.211618 | 2.466805 | 4.362033 | 2.165975 |
| 12:00 | 0 | 4.574689 | 0 | 3.162748 | 3.727524 | 2.4003 | 3.953435 | 2.315583 | 4.094629 | 2.033195 |
| 13:00 | 0 | 5.993776 | 0 | 4.143845 | 4.883817 | 3.144882 | 5.179806 | 3.033887 | 5.364799 | 2.6639 |
| 14:00 | 0 | 4.892116 | 0 | 3.382204 | 3.986169 | 2.566851 | 4.227755 | 2.476256 | 4.378746 | 2.174274 |
| 15:00 | 0 | 5.820124 | 0 | 4.02379 | 4.742324 | 3.053769 | 5.029737 | 2.945989 | 5.209371 | 2.586722 |
| 16:00 | 0 | 2.756017 | 0 | 1.905394 | 2.245643 | 1.446058 | 2.381743 | 1.395021 | 2.466805 | 1.224896 |
| 17:00 | 0 | 2.446058 | 0 | 1.691102 | 1.993084 | 1.283426 | 2.113877 | 1.238128 | 2.189373 | 1.087137 |
| 18:00 | 0 | 2.388174 | 0 | 1.651083 | 1.94592 | 1.253054 | 3.063854 | 1.208829 | 2.137563 | 2.061411 |
| 19:00 | 0 | 2.371369 | 0 | 1.639465 | 1.932227 | 1.244237 | 3.049331 | 1.200323 | 2.122522 | 2.053942 |
| 20:00 | 0 | 2.558091 | 0 | 1.768557 | 2.084371 | 1.342208 | 3.210696 | 1.294836 | 2.28965 | 2.136929 |
| 21:00 | 0 | 2.429253 | 0 | 1.679484 | 1.979391 | 1.274608 | 4.099355 | 1.229622 | 2.174331 | 2.079668 |
| 22:00 | 0 | 4.593361 | 0 | 3.175657 | 3.742739 | 2.410097 | 3.969571 | 2.325035 | 4.111342 | 2.041494 |
| 23:00 | 0 | 3.96888 | 0 | 2.052559 | 2.419087 | 1.557746 | 3.565698 | 1.502766 | 2.657331 | 1.819502 |
| 0:00 | 0 | 3.620539 | 0 | 2.503089 | 2.950069 | 1.899666 | 3.128861 | 1.832619 | 3.240606 | 1.609129 |

**Table A3.** Hourly load data for each bus of the IEEE test system (c).

| Time | Bus 21 | Bus 22 | Bus 23 | Bus 24 | Bus 25 | Bus 26 | Bus 27 | Bus 28 | Bus 29 | Bus 30 |
|---|---|---|---|---|---|---|---|---|---|---|
| 1:00 | 3.890041 | 0 | 1.417704 | 2.368603 | 0 | 1.469571 | 0 | 0 | 1.279391 | 2.697095 |
| 2:00 | 5.656639 | 0 | 1.332642 | 2.226487 | 0 | 1.381397 | 0 | 0 | 1.202628 | 2.53527 |
| 3:00 | 5.578838 | 0 | 1.304288 | 2.179115 | 0 | 1.352006 | 0 | 0 | 1.17704 | 2.481328 |
| 4:00 | 5.812241 | 0 | 1.38935 | 2.321231 | 0 | 1.44018 | 0 | 0 | 1.253804 | 2.643154 |
| 5:00 | 5.551141 | 0 | 2.203527 | 3.144917 | 0 | 2.254876 | 0 | 0 | 1.266598 | 2.670124 |
| 6:00 | 3.75778 | 0 | 1.369502 | 2.288071 | 0 | 1.419606 | 0 | 0 | 1.235892 | 2.605394 |
| 7:00 | 3.578838 | 0 | 1.304288 | 2.179115 | 0 | 1.352006 | 0 | 0 | 1.17704 | 2.481328 |
| 8:00 | 3.501037 | 0 | 1.275934 | 2.131743 | 0 | 1.322614 | 0 | 0 | 1.151452 | 2.427386 |
| 9:00 | 4.045643 | 0 | 1.474412 | 2.463347 | 0 | 1.528354 | 0 | 0 | 1.330567 | 2.804979 |
| 10:00 | 5.005187 | 0 | 1.824112 | 3.047603 | 0 | 1.890848 | 0 | 0 | 1.64615 | 3.470263 |
| 11:00 | 6.768672 | 0 | 2.466805 | 4.121369 | 0 | 2.557054 | 0 | 0 | 2.226141 | 4.692946 |
| 12:00 | 6.353734 | 0 | 2.315583 | 3.868718 | 0 | 2.4003 | 0 | 0 | 2.089673 | 4.405256 |
| 13:00 | 8.324689 | 0 | 3.033887 | 5.068811 | 0 | 3.144882 | 0 | 0 | 2.737898 | 5.771784 |
| 14:00 | 6.794606 | 0 | 2.476256 | 4.13716 | 0 | 2.566851 | 0 | 0 | 2.23467 | 4.710927 |
| 15:00 | 8.083506 | 0 | 2.945989 | 4.921957 | 0 | 3.053769 | 0 | 0 | 2.658575 | 5.604564 |
| 16:00 | 3.827801 | 0 | 1.395021 | 2.330705 | 0 | 1.446058 | 0 | 0 | 1.258921 | 2.653942 |
| 17:00 | 3.397303 | 0 | 1.238128 | 2.06858 | 0 | 1.283426 | 0 | 0 | 1.117335 | 2.355463 |
| 18:00 | 6.316909 | 0 | 1.208829 | 2.019629 | 0 | 1.253054 | 0 | 0 | 1.090894 | 2.299723 |
| 19:00 | 6.293568 | 0 | 1.200323 | 2.005417 | 0 | 1.244237 | 0 | 0 | 1.083218 | 2.283541 |
| 20:00 | 6.552905 | 0 | 1.294836 | 2.163324 | 0 | 1.342208 | 0 | 0 | 1.168511 | 2.463347 |
| 21:00 | 6.373963 | 0 | 1.229622 | 2.054368 | 0 | 1.274608 | 0 | 0 | 1.109659 | 2.339281 |
| 22:00 | 6.379668 | 0 | 2.325035 | 3.884509 | 0 | 2.410097 | 0 | 0 | 2.098202 | 4.423237 |
| 23:00 | 5.123444 | 0 | 2.502766 | 3.510719 | 0 | 1.957746 | 0 | 0 | 1.856155 | 3.858921 |
| 0:00 | 5.028527 | 0 | 1.832619 | 3.061814 | 0 | 1.899666 | 0 | 0 | 1.653827 | 3.486445 |

**Table A4.** The output range of 3WPPs per hour.

| Time | WPP1 | | WPP2 | | WPP3 | |
|---|---|---|---|---|---|---|
| | Lower | Upper | Lower | Upper | Lower | Upper |
| 1:00 | 0.14 | 5.07 | 0.12 | 4.38 | 0.09 | 3.34 |
| 2:00 | 4.79 | 7.62 | 4.15 | 6.59 | 3.16 | 5.02 |
| 3:00 | 6.08 | 10.96 | 5.26 | 9.48 | 4.00 | 7.22 |
| 4:00 | 8.06 | 10.38 | 6.97 | 8.98 | 5.31 | 6.84 |
| 5:00 | 8.37 | 9.44 | 7.24 | 8.16 | 5.51 | 6.22 |
| 6:00 | 2.98 | 8.48 | 2.57 | 7.34 | 1.96 | 5.59 |
| 7:00 | 3.61 | 4.53 | 3.12 | 3.92 | 2.38 | 2.98 |
| 8:00 | 2.73 | 4.63 | 2.36 | 4.00 | 1.80 | 3.05 |
| 9:00 | 0.33 | 1.86 | 0.00 | 1.61 | 0.22 | 1.23 |
| 10:00 | 0.00 | 2.04 | 0.00 | 1.76 | 0.00 | 1.34 |
| 11:00 | 0.00 | 1.37 | 0.00 | 1.19 | 0.00 | 0.90 |
| 12:00 | 0.00 | 1.27 | 0.00 | 1.10 | 0.00 | 0.84 |
| 13:00 | 0.00 | 1.17 | 0.00 | 1.01 | 0.00 | 0.77 |
| 14:00 | 0.00 | 2.43 | 0.00 | 2.11 | 0.00 | 1.60 |
| 15:00 | 0.00 | 2.62 | 0.00 | 2.26 | 0.00 | 1.72 |
| 16:00 | 0.84 | 3.49 | 0.73 | 3.02 | 0.00 | 2.30 |
| 17:00 | 1.97 | 6.02 | 1.70 | 5.21 | 1.30 | 3.96 |
| 18:00 | 7.42 | 12.05 | 6.42 | 10.43 | 4.89 | 7.94 |
| 19:00 | 6.85 | 11.20 | 5.93 | 9.69 | 4.52 | 7.38 |
| 20:00 | 12.11 | 14.58 | 10.48 | 12.62 | 7.98 | 9.61 |
| 21:00 | 11.25 | 14.83 | 9.73 | 12.83 | 7.41 | 9.77 |
| 22:00 | 9.09 | 14.96 | 7.87 | 12.94 | 5.99 | 9.86 |
| 23:00 | 9.01 | 14.98 | 7.79 | 12.96 | 5.94 | 9.87 |
| 0:00 | 10.04 | 14.98 | 8.69 | 12.94 | 6.62 | 9.67 |

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
