# Peer review of "Day-Ahead Scheduling Model of the Distributed Small Hydro-Wind-Energy Storage Power System Based on Two-Stage Stochastic Robust Optimization"

_sustainability, doi:10.3390/su11102829_

Reviewer 1 Report

This work proposed a a two-stage stochastic robust optimization approach for day-ahead market operation dispatching of the distributed power system with high renewable penetration that consists

of SHPPs, WPPs, BESSs, and TPPs.

An IEEE 30-bus test system with three SHPPs, three WPPs, two TPPs and two BESSs is adopted as an example.

Some more detailed comments are given below. I hope that if the authors will take them into account the paper will be improved.

1. A simple numerical example is needed to provide an illustration of the Figure 3 in Section 3.4.

2. The website of detailed data in section 4.2 can list in the reference.

3. Provide all algorithmic parameter setting of the FCM-CCQ method, CCG algorithm and AD algorithm in Section 4.

4. Comparison of proposed methodology with exiting techniques/its benefits. Please add a detailed discussion.

Author Response

Point 1: A simple numerical example is needed to provide an illustration of the Figure 3 in Section 3.4.

Response 1: Maybe the illustration of the Figure 3 of previous paper was not clear enough. The whole section 4 is a numerical example which applies the model mentioned in Figure 3. To better illustrate all this, an explanatory paragraph is added at the beginning of chapter 4 so that you can better understand the application scenarios of the model in Figure 3.

Firstly, the setting of the general basic background should be completed. In terms of whether the uncertainty of wind power output is considered, it is divided into two different cases to form two problems, namely, deterministic optimization and uncertain optimization. Deterministic optimization is a general linear integer programming problem, which can be solved directly. Uncertainty optimization is solved by two-stage stochastic robust optimization as shown in Figure. 3. The solution idea can be roughly divided into two parts. In the first part, the FCM-CCQ method is used for clustering analysis of the inflow scenario of small hydropower (corresponding to the process in the left half of Figure 3). In this numerical example, 100 groups of real historical water inflow data of small hydropower are input, and the final clustering result is determined by FCM-CCQ algorithm. In the second part, based on each type of incoming water scenario, the two-stage robust optimization method (corresponding to the process in the right half of Figure 3) is used to identify the scenario with the greatest negative impact on the power system according to the uncertainty interval of the wind power output. On this basis, the optimal scheduling strategy is obtained. This strategy can be understood as that no matter how the actual output of wind power changes in the future, the total cost of system operation will not exceed the total cost under the optimal strategy. In the end, the system operation cost calculated under the optimal strategy under each kind of small hydropower inflow scenario is multiplied by the corresponding occurrence probability of this category, and then the expected operation cost of the system is obtained by summing them.

Point 2: The website of detailed data in section 4.2 can list in the reference.

Response 2: The website of detailed data has been moved to the reference.

Point 3: Provide all algorithmic parameter setting of the FCM-CCQ method, CCG algorithm and AD algorithm in Section 4.

Response 3: The parameters that FCM needs to input in this paper are the inflow data of 100 groups of small hydropower and the number of the final classification. The CCQ algorithm is an evaluation model used to assess the quality of classification, so the input parameter of CCQ is just the result of each classification which obtained from FCM.

Besides, CCG and AD algorithm are the algorithm to solve the max-min programming problem, their algorithmic parameter setting are in power system parameter setting in section 4.2.

Point 4: Comparison of proposed methodology with exiting techniques/its benefits. Please add a detailed discussion.

Response 4:

(1)     Generally speaking, the ordinary clustering analysis methods such as K-means and the FCM, will be given in advance to aggregate number of categories. For example, the 100 scenario data clustering to 20 categories, of course, it can use general clustering method, but cannot explain why is divided into 20 categories. Is this clustering scheme the best one? These questions cannot be answered by ordinary clustering method. However, CCQ algorithm which is adopted to evaluate each classification result can solve this issue and select the best scheme via indicators calculation. Therefore, the conventional clustering method combined with CCQ algorithm is better than only using the general clustering method.

(2)     Compared with deterministic optimization, two-stage robust optimization is adopted to consider the impact of real-time output change of wind power on the power system. Deterministic optimization is based on the expected output value of wind power, and the actual generation process will have deviation, which will affect the stability and reliability of the power system. However,robust optimization can deal with this problem well, reduce the risk of the whole system, and ensure the safe and stable operation of the system.

Reviewer 2 Report

This paper presents a scheduling model of a distributed small hydroelectric energy storage energy system based on stochastic optimization. The authors show a two-stage stochastic robust optimization approach to smooth out hydro power and wind power output fluctuation with the aim of minimizing the total expected system operation cost under multiple cluster water inflow scenarios and the worst case of wind power output. The abstract does not clearly present the description of the study presented, nor does it indicate the contributions made by the authors to the topic addressed. This summary does not include the main contributions and results obtained.

The introduction includes the state-of-the-art of the problem studied. It would be advisable to increase the number of bibliographic references; as well as indicate the differences among the research works mentioned. In the same way it would be advisable to indicate the advantages and disadvantages of the different referenced studies. It may also be advisable to develop a comparison between the algorithm applied in each case. It is recommended that the authors clearly indicate their contribution to the study with respect to other researches develops and cited in the references.

Section 2 shows a description of the problem addressed. While section 3 shows the basic principle of operation and the theoretical concepts. The mathematical equations shown also facilitate the follow-up of the concepts. It would be advisable to explain how the SOC charge and discharge states of the batteries are determined, as well as their efficiency; see section 3.2.2. Please explain in more detail the reformulation of the model as well as its constraints.

Section 4 shows different case studies and the results obtained. The references are current. All of them have been published in this last decade. It would be advisable to carry out a search of more extensive academic work. all of them related to the topic addressed.

Review the text in English, there are different words and terms that are not used frequently.

Author Response

Thank you very much for your detailed and specific Suggestions! I have made specific modifications to each of them.

Point 1: This paper presents a scheduling model of a distributed small hydroelectric energy storage energy system based on stochastic optimization. The authors show a two-stage stochastic robust optimization approach to smooth out hydro power and wind power output fluctuation with the aim of minimizing the total expected system operation cost under multiple cluster water inflow scenarios and the worst case of wind power output. The abstract does not clearly present the description of the study presented, nor does it indicate the contributions made by the authors to the topic addressed. This summary does not include the main contributions and results obtained.

Response 1: The abstract has been revised according to the suggestions. The details are as follows:

Abstract: With renewable energy sources (RESs) highly penetrating into the power system, new problems emerge for the independent system operator (ISO) to maintenance and keep the power system safe and reliable in the day-ahead dispatching process under the fluctuation caused by renewable energy. In this paper, considering the small hydropower with no reservoir different from the other hydro optimization research and wind power uncertain circumstances, a day-ahead scheduling model is proposed for a distributed power grid system which contains several distributed generators, such as small hydropower and wind power, and energy storage systems. To solve this model, a two-stage stochastic robust optimization approach is presented to smooth out hydro power and wind power output fluctuation with the aim of minimizing the total expected system operation cost under multiple cluster water inflow scenarios and the worst case of wind power output uncertainty. More specifically, before dispatching and clearing, it is necessary to cluster the historical inflow scenarios of small hydropower into several typical scenarios via the Fuzzy C-means (FCM) clustering method, and the clustering comprehensive quality (CCQ) method is presented to evaluate whether these scenarios are representative, which is previously ignored by cluster research. It can be found through numerical examples that FCM-CCQ can explain the classification more reasonably than the common clustering method. Then, optimizing the two stages scheduling which contain pre-clearing stage and rescheduling stage under each typical inflow scenarios after clustering, and calculating the final operating cost under the worst wind power output scenario. To conduct the proposed model, the day-ahead scheduling procedure on the IEEE 30-bus test system is simulated with real hydropower and wind power data. Compared with traditional deterministic optimization, the results of two-stage stochastic robust optimization structured in this paper increase the total cost of the system, but enhance the conservative scheduling strategy, improve the stability and reliability of power system, and reduce the risk of decision-making simultaneously.

Point 2: The introduction includes the state-of-the-art of the problem studied. It would be advisable to increase the number of bibliographic references; as well as indicate the differences among the research works mentioned. In the same way it would be advisable to indicate the advantages and disadvantages of the different referenced studies. It may also be advisable to develop a comparison between the algorithm applied in each case. It is recommended that the authors clearly indicate their contribution to the study with respect to other researches develops and cited in the references.

Response 2: Compared with other studies, the important contributions made in this paper have been revised on the basis of the original manuscript. For details, please refer to the revised draft.

Point 3: Section 2 shows a description of the problem addressed. While section 3 shows the basic principle of operation and the theoretical concepts. The mathematical equations shown also facilitate the follow-up of the concepts. It would be advisable to explain how the SOC charge and discharge states of the batteries are determined, as well as their efficiency; see section 3.2.2. Please explain in more detail the reformulation of the model as well as its constraints.

Response 3: State-of-charge of the batteries                                                are modelled as follow:

 represents energy stored in BESS ;  represents rated energy capacity.

 ,  refer to charge/ discharge efficiency respectively, and the default charge/discharge efficiency are constant.

Because the research of reformulation of two-stage robust optimization has been very mature. Therefore, this paper does not describe the reformulation in detail. For more detail derivation process, the following references can help you understand it comprehensively. Both of them are in references.

[1]     Zeng B, Zhao L. Solving two-stage robust optimization problems using a column-and-constraint generation method[J]. Operations Research Letters, 2013, 41(5): 457-461.

[2]     Bertsimas D, Litvinov E, Sun X A, et al. Adaptive robust optimization for the security constrained unit commitment problem[J]. IEEE transactions on power systems, 2013, 28(1): 52-63.

Point 4: Section 4 shows different case studies and the results obtained. The references are current. All of them have been published in this last decade. It would be advisable to carry out a search of more extensive academic work. all of them related to the topic addressed.

Response 4: References have been updated to include some recent studies.

Point 5: Review the text in English, there are different words and terms that are not used frequently.

Response 5: I’m so sorry that I’m not sure which words and terms you suggest should be replaced.

Reviewer 3 Report

Revise the whole paper based on the comments:

See the paper: A multi-objective optimization of energy, economic, and carbon emission in a production model under sustainable supply chain management, Applied Sciences 8 (10), 1744 and make the author contribution and finally show the novelty like this.

The abstract should be rewritten based on the findings of the research.

The model should be explained properly with proper assumptions and proper references. Atleast 3 references for each idea should be given.

New references from recent publication should be added.

The data source should be given to validate the model.

The conclusions and managerial insights should be added perfectly for this research.

Mention properly the future extension of the model.

More recent reference should be considered like as Effect of energy and failure rate in a multi-item smart production system, Energies 11 (11), 1-21

Author Response

Point 1: See the paper: A multi-objective optimization of energy, economic, and carbon emission in a production model under sustainable supply chain management, Applied Sciences 8 (10), 1744 and make the author contribution and finally show the novelty like this.

Response 1:

Author contribution: J.D. supervised the proposed research work, whereas P.Y. conceptualized the proposed research work, collected the data, programmed and analysed the methodology, and written the original draft. S.N. helped in review writing and editing with validation of research work.

Novelty:

Compared with the previous research of distributed power system dispatching, the main contribution is that a two-stage stochastic robust optimization model is proposed, which applied to solve the day-ahead dispatching problem with thermal power wind power run-of-river SHPPs and energy storage device. Moreover, considering the uncertainty of small hydropower, a FCM-CCQ algorithm is proposed which is adopted to cluster the historical upstream inflow data of small hydropower quickly and effectively. This research is providing a reference for renewable energy consumption and ISO can fully consider the interference of uncertain factors and formulate an optimal power generation dispatch plan, thereby reducing the economic cost of the system. Broadly speaking, this work can effectively reduce the carbon dioxide emissions of power systems and mitigate global warming while promoting distributed energy consumption.

Point 2: The abstract should be rewritten based on the findings of the research.

Response 2: The abstract has been rewritten based on the findings of the research.

Abstract: With renewable energy sources (RESs) highly penetrating into the power system, new problems emerge for the independent system operator (ISO) to maintenance and keep the power system safe and reliable in the day-ahead dispatching process under the fluctuation caused by renewable energy. In this paper, considering the small hydropower with no reservoir different from the other hydro optimization research and wind power uncertain circumstances, a day-ahead scheduling model is proposed for a distributed power grid system which contains several distributed generators, such as small hydropower and wind power, and energy storage systems. To solve this model, a two-stage stochastic robust optimization approach is presented to smooth out hydro power and wind power output fluctuation with the aim of minimizing the total expected system operation cost under multiple cluster water inflow scenarios and the worst case of wind power output uncertainty. More specifically, before dispatching and clearing, it is necessary to cluster the historical inflow scenarios of small hydropower into several typical scenarios via the Fuzzy C-means (FCM) clustering method, and the clustering comprehensive quality (CCQ) method is presented to evaluate whether these scenarios are representative, which is previously ignored by cluster research. It can be found through numerical examples that FCM-CCQ can explain the classification more reasonably than the common clustering method. Then, optimizing the two stages scheduling which contain pre-clearing stage and rescheduling stage under each typical inflow scenarios after clustering, and calculating the final operating cost under the worst wind power output scenario. To conduct the proposed model, the day-ahead scheduling procedure on the IEEE 30-bus test system is simulated with real hydropower and wind power data. Compared with traditional deterministic optimization, the results of two-stage stochastic robust optimization structured in this paper increase the total cost of the system, but enhance the conservative scheduling strategy, improve the stability and reliability of power system, and reduce the risk of decision-making simultaneously.

Point 3: The model should be explained properly with proper assumptions and proper references. Atleast 3 references for each idea should be given.

Response 3: The problem assumptions have been added in Section 3.1, more proper references for the model have been increased.

Point 4: New references from recent publication should be added.

Response 4: New references from recent publication have been added.

Point 5: The data source should be given to validate the model.

Response 5: The data source has been given in Section 4.2.

Point 6: The conclusions and managerial insights should be added perfectly for this research.

Response 6: The conclusions have been revised, and add the managerial insights.

Point 7: Mention properly the future extension of the model.

Response 7: The future extension of the model has been added in the conclusion.

Point 8: More recent reference should be considered like as Effect of energy and failure rate in a multi-item smart production system, Energies 11 (11), 1-21.

Response 8: More recent reference have been considered and added in the references.

Round  2

Reviewer 1 Report

The authors have carefully addressed the previous comments of the reviewer and significantly improved the manuscript. 

Author Response

Point 1: The authors have carefully addressed the previous comments of the reviewer and significantly improved the manuscript.

Thank you very much for taking the time to put forward so many valuable comments.

Reviewer 3 Report

The authors did not follow my last instructions all. Those should be resolved first. 

Author Response

Thank you very much for your detailed and specific suggestions! I revised the manuscript again according to the comments of the reviewer.

Point 1: See the paper: A multi-objective optimization of energy, economic, and carbon emission in a production model under sustainable supply chain management, Applied Sciences 8 (10), 1744 and make the author contribution and finally show the novelty like this

Response 1:

The author contribution and the novelty have been revised according to the paper mentioned by reviewer, and add this paper to the references. The specific modifications are as follows:

Author contribution: J.D. supervised the proposed research work, whereas P.Y. conceptualized the proposed research work, collected the data, programmed and analysed the methodology, and written the original draft. S.N. helped in review writing and editing with validation of research work.

Novelty:

Compared with the previous research of distributed power system dispatching, the main contribution is that a two-stage stochastic robust optimization model is proposed, which applied to solve the day-ahead dispatching problem with thermal power wind power run-of-river SHPPs and energy storage device. Moreover, considering the uncertainty of small hydropower, a FCM-CCQ algorithm is proposed which is adopted to cluster the historical upstream inflow data of small hydropower quickly and effectively. This research is providing a reference for renewable energy consumption and ISO can fully consider the interference of uncertain factors and formulate an optimal power generation dispatch plan, thereby reducing the economic cost of the system. Broadly speaking, this work can effectively reduce the carbon dioxide emissions of power systems and mitigate global warming while promoting distributed energy consumption.

Point 2: The abstract should be rewritten based on the findings of the research.

Response 2: The abstract has been rewritten based on the findings of the research.

Abstract: With renewable energy sources (RESs) highly penetrating into the power system, new problems emerge for the independent system operator (ISO) to maintenance and keep the power system safe and reliable in the day-ahead dispatching process under the fluctuation caused by renewable energy. In this paper, considering the small hydropower with no reservoir different from the other hydro optimization research and wind power uncertain circumstances, a day-ahead scheduling model is proposed for a distributed power grid system which contains several distributed generators, such as small hydropower and wind power, and energy storage systems. To solve this model, a two-stage stochastic robust optimization approach is presented to smooth out hydro power and wind power output fluctuation with the aim of minimizing the total expected system operation cost under multiple cluster water inflow scenarios and the worst case of wind power output uncertainty. More specifically, before dispatching and clearing, it is necessary to cluster the historical inflow scenarios of small hydropower into several typical scenarios via the Fuzzy C-means (FCM) clustering method, and the clustering comprehensive quality (CCQ) method is presented to evaluate whether these scenarios are representative, which is previously ignored by cluster research. It can be found through numerical examples that FCM-CCQ can explain the classification more reasonably than the common clustering method. Then, optimizing the two stages scheduling which contain pre-clearing stage and rescheduling stage under each typical inflow scenarios after clustering, and calculating the final operating cost under the worst wind power output scenario. To conduct the proposed model, the day-ahead scheduling procedure on the IEEE 30-bus test system is simulated with real hydropower and wind power data. Compared with traditional deterministic optimization, the results of two-stage stochastic robust optimization structured in this paper increase the total cost of the system, but enhance the conservative scheduling strategy, improve the stability and reliability of power system, and reduce the risk of decision-making simultaneously.

Point 3: The model should be explained properly with proper assumptions and proper references. Atleast 3 references for each idea should be given.

Response 3: The problem assumptions have been added in Section 3.1, more proper references for the model have been increased.

Point 4: New references from recent publication should be added.

Response 4: New references from recent publication have been added. See the paper's references for details.

Point 5: The data source should be given to validate the model.

Response 5: The data source has been given in Section 4.2. These relevant parameters are collected from Refs. [12], [28], [29].

Point 6: The conclusions and managerial insights should be added perfectly for this research.

Response 6: The conclusions have been revised, and add the managerial insights.

Point 7: Mention properly the future extension of the model.

Response 7: The future extension of the model has been added in the conclusion. The specific modifications are as follows:

Furthermore, the proposed scheduling model adopts a conservative scheduling strategy to avoid the adverse impact of intermittency and randomness of renewable energy generation on the power system to some extent, and at the same time promotes the absorption of distributed renewable energy. Paving the way for large-scale absorption of renewable energy and sustainable development of the society in the future.

Finally, the model established in this paper is relatively simple and ideal, and but it is hoped that the work can provide useful reference for renewable energy consumption and day-ahead dispatching operation of distributed power system. In addition, further research can be carried out in the direction of more complex distributed system with the real-time change of power load demand and the response of customer demands in the future.

Point 8: More recent reference should be considered like as Effect of energy and failure rate in a multi-item smart production system, Energies 11 (11), 1-21.

Response 8: The reference mentioned by reviewer has been added to the references.

Round  3

Reviewer 3 Report

The paper can be accepted for publication.